# SNIP1 and PRC2 coordinate cell fates of neural progenitors during brain development

Yurika Matsui[1], Mohamed Nadhir Djekidel [2], Katherine Lindsay[1], Parimal Samir[1,3], Nina Connolly[1], Gang Wu [2], Xiaoyang Yang [1], Yiping Fan[2], Beisi Xu [2] & Jamy C. Peng [1] ✉

Stem cell survival versus death is a developmentally programmed process essential for morphogenesis, sizing, and quality control of genome integrity and cell fates. Cell death is pervasive during development, but its programming is little known. Here, we report that Smad nuclear interacting protein 1 (SNIP1) promotes neural progenitor cell survival and neurogenesis and is, therefore, integral to brain development. The SNIP1-depleted brain exhibits dysplasia with robust induction of caspase 9-dependent apoptosis. Mechanistically, SNIP1 regulates target genes that promote cell survival and neurogenesis, and its activities are influenced by TGFβ and NFκB signaling pathways. Further, SNIP1 facilitates the genomic occupancy of Polycomb complex PRC2 and instructs H3K27me3 turnover at target genes. Depletion of PRC2 is sufficient to reduce apoptosis and brain dysplasia and to partially restore genetic programs in the SNIP1-depleted brain in vivo. These findings suggest a loci-specific regulation of PRC2 and H3K27 marks to toggle cell survival and death in the developing brain.

Neural progenitor cells (NPCs) are self-renewing and multipotent cells that give rise to neurons, oligodendrocytes, and astrocytes in the central nervous system (CNS). The precise size and structural organization of the brain is achieved by the exquisite balance between NPC division, death, and differentiation. Cell death is prominent in the normal developing brain especially in the proliferative zones of the cortex where NPCs reside[1–4]. Multiple models have been proposed to explain the high degree of NPC death, including DNA damage due to accumulative replicative and transcriptional burdens, insufficient supply of neurotrophic factors, and self-destroying genetic programs that are temporally and spatially regulated during development[4–8]. However, how NPCs execute cell death during embryonic development remains relatively little known.

In contrast, the cell-intrinsic control of NPC division and differentiation is better understood and depends on the Polycomb repressive complex 2 (PRC2). PRC2 deposits H3K27 trimethylation (H3K27me3) and suppresses RNA polymerase II–dependent transcription[9–12]. PRC2 plays essential roles throughout CNS development, such as neural tube closure, expansion of NPCs, and cell fate specification[13–15]. Additionally, the involvement of PRC2-mediated H3K27me3 is implicated in neuroprotection[9,16,17]. Aberrant increases of H3K27me3 and higher stability of the PRC2 enzymatic subunit Enhancer of zeste 2 (EZH2) are observed in ataxia-telangiectasia[16], whereas H3K27me3 reduction is linked to Parkinson's disease and Huntington's disease[9,17]. PRC2 and H3K27me3 suppress gene targets involved in neurodegeneration in the adult brain[9,16].

[1]Department of Developmental Neurobiology, St. Jude Children's Research Hospital, 262 Danny Thomas Place, Memphis, TN 38105, USA. [2]Center for Applied Bioinformatics, St. Jude Children's Research Hospital, 262 Danny Thomas Place, Memphis, TN 38105, USA. [3]Present address: Department of Microbiology and Immunology, University of Texas Medical Branch, 301 University Blvd, Medical Research Building, Room 7, 138E, Galveston, TX 77550, USA. ✉e-mail: jamy.peng@stjude.org

To better understand PRC2 function and regulation in brain development, we performed proteomic analysis of PRC2 co-immunoprecipitates[18]. This analysis uncovered SNIP1 as a putative PRC2 interactor, which is consistent with other proteomic analyses of PRC2 purification[19,20]. SNIP1 was initially discovered as an interactor of SMAD proteins[21]. The N-terminus of SNIP1 binds to histone acetyl-transferases p300/CBP at their CH1 domain to disrupt the SMAD4–p300/CBP complex formation and dampen BMP/TGFβ signaling[22–25]. Using a similar strategy, SNIP1 can also dampen NFκB signaling by disrupting RELA/p65-p300/CBP complex formation[26–30]. Besides its role in TGFβ or NFκB signaling, SNIP1 acts on oncogenes or tumor suppressors to promote cell transformation and cell cycle. In cancer cells, SNIP1 can increase the stability of c-MYC and its complex with p300/CBP[31], disrupt the RB-HDAC1 complex[32], or promote cyclin D1 expression[33,34]. Ectopic expression of SNIP1 induces defective patterning in the *Xenopus* embryos[21]. Global knockout (KO) of Snip1 in zebrafish embryos causes reduction in GABAergic and glutamatergic neurons[35]. Mutations in *SNIP1* has also been linked to neurodevelopmental disorders. The E366G variant of *SNIP1* has been linked to patients with psychomotor retardation, epilepsy, and craniofacial dysmorphism[36,37]. Additionally, we examined publicly available data from Human Gene Mutation Database[38] and Mastermind[39] and identified a *SNIP1* variant, R111C, that is significantly associated with epilepsy and skull dysplasia (Supplementary Fig. 1a). These data suggest a critical role of SNIP1 in human neurodevelopment.

Using *Snip1* conditional deletion in the mouse embryonic brain, here we show that SNIP1 is required for suppressing apoptotic genes while promoting neurogenic genes. Mechanistically, genomic occupancy of SNIP1 is influenced by TGFβ and NFκB signaling pathways. At the target genes, SNIP1 binds to PRC2 and promotes genomic occupancy of PRC2 and H3K27me3 deposition. The depletion of a PRC2 subunit Embryonic ectoderm development (EED) partially restores gene expression, NPC function, and development in the SNIP1-depleted brain. Our study reveals an epigenetic pathway that is essential for coordinating survival, self-renewal, and differentiation of stem cells in the developing brain.

## Results

### SNIP1 is required for the survival of NPCs in the murine embryonic brain

We first examined the expression of *Snip1* in the murine embryonic brain by RNAscope[40]. At embryonic day E11.5 and E13.5, *Snip1* transcripts were expressed in nearly all cells and robustly expressed in the neuroepithelia lining the ventricles, where NPCs reside (Supplementary Fig. 1b, c). To study SNIP1 in the murine embryonic brain, we used *Nestin* (*Nes*)::*Cre* to conditionally deplete SNIP1 in NPCs, hereafter referred to as *Snip1^Nes*-KO. *Nes*::Cre is expressed in NPCs to recombine the flox sites and excise exon 2 of *Snip1* (Fig. 1a, Supplementary Fig. 1d–f, Supplementary Fig. 2a–d). By E15, *Snip1^Nes*-KO embryos displayed severe thinning of brain tissues and dysplasia with 100% penetrance (Fig. 1b, c, *p* < 0.0001 by Fisher's exact test). To understand the cellular underpinnings of the brain dysplasia in *Snip1^Nes*-KO, we examined cell proliferation and apoptosis of NPCs. NPCs were identified by their expression of the neural stem cell marker SRY-box 2 (SOX2). To identify proliferating cells in vivo, we injected BrdU into pregnant dams and/or detected the proliferative marker Ki67. Quantification of these markers in neuroepithelia did not reveal significant difference in proliferative NPCs in sibling control and *Snip1^Nes*-KO embryos (Supplementary Fig. 2e–h).

We next probed apoptosis by IF of cleaved (cl)-caspase 3. At E13.5, all ventricles of *Snip1^Nes*-KO displayed strong induction of cl-caspase 3 in the subventricular zones of neuroepithelia (Fig. 1d–f). The cl-caspase 3 signals overlapped SOX2-positive NPCs and TBR2-positive intermediate progenitors (Supplementary Fig. 2i–l), which were markedly reduced in *Snip1^Nes*-KO neuroepithelia (Fig. 1d–i). INSM1-positive

intermediate progenitors were similarly reduced in *Snip1^Nes*-KO midbrain (Fig. 1j, k). These data suggest that SNIP1 suppresses apoptosis in NPCs and intermediate progenitors in the developing brain.

To capture the earlier events of the *Snip1^Nes*-KO brain, we examined cl-caspase 3 in the E11.5 embryonic brain. By E11.5, cl-caspase 3 signals were detected throughout the *Snip1^Nes*-KO brain (Supplementary Fig. 3a–d). As *Nes*::Cre is turned on by E10.5[41], SNIP1 depletion likely induces apoptosis within 24 h. The quantification of SOX2-positive cells in lateral, third, and fourth ventricles showed that only NPCs in the fourth ventricle became depleted by E11.5 (Supplementary Fig. 3e). Hereafter, we focused our analyses to lateral and third ventricles to study apoptosis control by SNIP1.

To ascertain that SNIP1 depletion consistently causes the observed apoptosis, we used *Emx1*::Cre to conditionally deplete SNIP1 in NPCs of the dorsal telencephalon[42]. The *Snip1^Emx1*-KO embryos also showed strong induction of apoptosis, loss of TBR2-positive intermediate progenitors, and dysplasia of the forebrain (Supplementary Fig. 3f–k). These findings support that in the developing murine brain, apoptosis induction and dysplasia are specific to SNIP1 depletion. These data further support our conclusion that SNIP1 is required for an anti-apoptotic and pro-survival mechanism in NPCs and intermediate progenitors.

### SNIP1 suppresses intrinsic apoptosis program

To shed light on the molecular underpinnings of the defective *Snip1^Nes*-KO NPCs, we performed RNA-sequencing (RNA-seq) of SOX2-positive NPCs sorted from E13.5 *Snip1^Nes*-KO and sibling controls (Fig. 2a, Supplementary Fig. 4a). We analyzed genes with count per million (CPM) values > 1 in either control or *Snip1^Nes*-KO NPCs. Using the criteria of false discovery rate (FDR) < 0.05 to compare 4-replicate datasets each from control and *Snip1^Nes-KO* NPCs, we identified 1210 upregulated genes and 1621 downregulated genes in *Snip1^Nes*-KO (Fig. 2b, Supplementary Fig. 4b–d).

Gene set enrichment analysis (GSEA) revealed that upregulated genes in *Snip1^Nes*-KO NPCs were enriched in functions related to p53-mediated apoptosis, H3K27me3 or bivalent promoters in NPCs and the brain, midbrain markers, spliceosomal small nuclear ribonucleoprotein particles, and signaling pathways involving TNF, IGF, TGFβ, and Hedgehog (Fig. 2c). As intrinsic and extrinsic apoptotic signatures were enriched in the upregulated genes in *Snip1^Nes*-KO NPCs (Fig. 2d, e), we examined the activation of these two pathways. IF showed little signal of cl-caspase 8, an effector of the extrinsic apoptosis pathway[43,44] (Supplementary Fig. 5a) but strong signals of cl-caspase 9, an effector of the intrinsic apoptosis pathway[45–47], throughout the neuroepithelia of ventricles (Fig. 2f). Additionally, quantification of cl-caspase 3 by FACS of FAM-DEVD-FMK staining[48,49] show that whereas inhibition of caspase 8 modestly altered apoptosis, inhibition of caspase 9 robustly reduced apoptosis in the *Snip1^Nes*-KO NPCs (Fig. 2g, h, Supplementary Fig. 4e, Supplementary Fig. 5b). A caspase 9 inhibitor, Z-LEHD-FMK, had effectively inhibited caspase 9 at 1 µM and caused cytotoxicity at 20 and 40 µM (Fig. 2h, Supplementary Fig. 4f). For identifying a cause of increased apoptosis, we did not detect an increase in DNA damage (Supplementary Fig. 5c–f) but detected strong increases of p53 signals in SOX2-positive NPCs (Supplementary Fig. 5g–i). These data suggest that the *Snip1^Nes*-KO embryo displayed dysregulated control of p53-mediated intrinsic apoptosis in NPCs. We propose that SNIP1 primarily suppresses intrinsic apoptosis as part of a neurodevelopmental program.

Downregulated genes in *Snip1^Nes*-KO NPCs were enriched in functions related to forebrain and cortex development, CNS neuron differentiation, chromosome segregation, NPC proliferation, axonogenesis, replication fork, and signaling pathways involving TLR and Rho (Fig. 2i). Although the *Snip1^Nes*-KO forebrain tissues displayed severe thinning as a consequence of apoptosis, the forebrain marker FOXG1 and the mid/hindbrain marker OTX2 were similarly detected

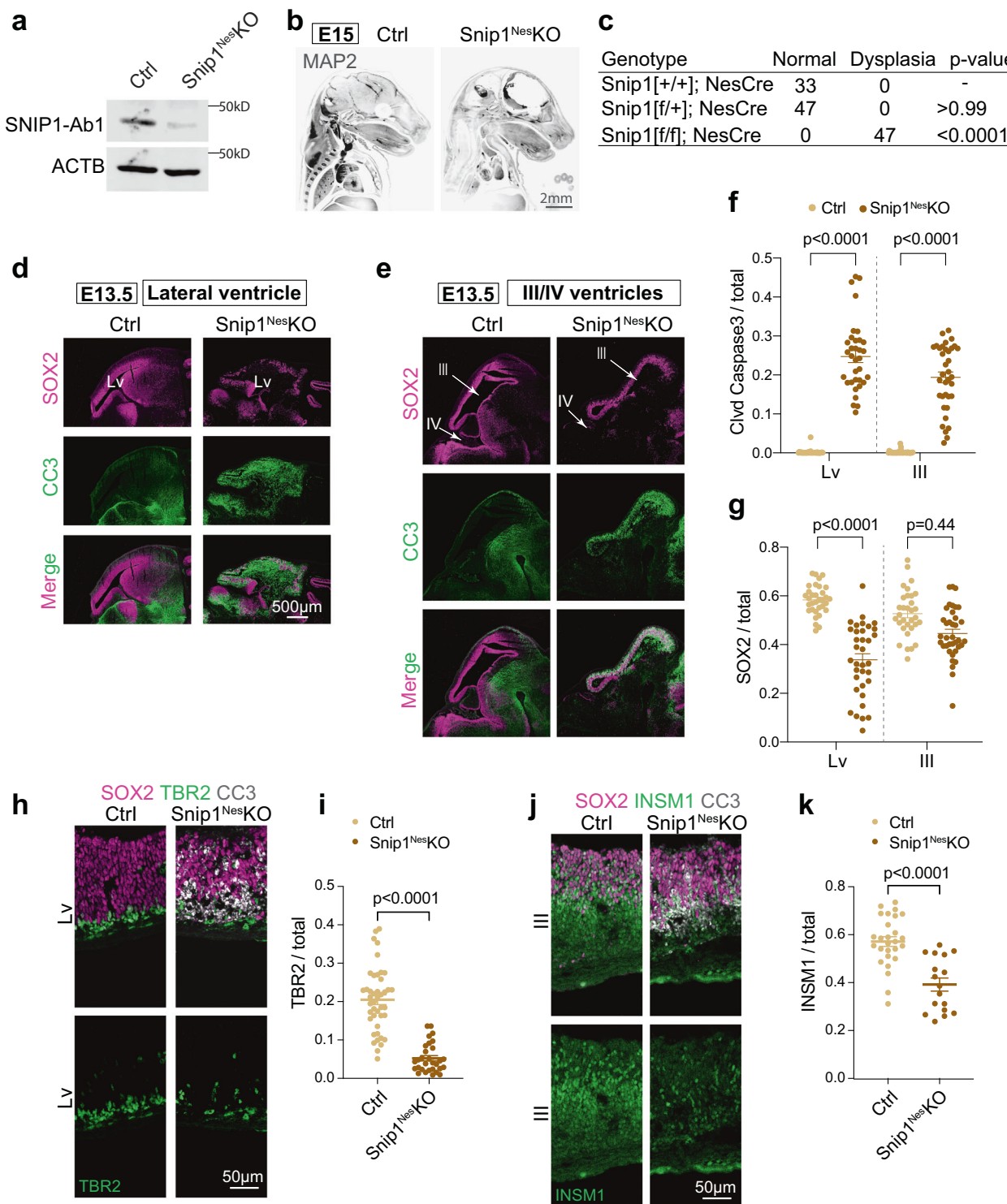

between the control and *Snip1*<sup>Nes</sup>-KO brains (Supplementary Fig. 5j). These data suggest that SNIP1 depletion did not alter forebrain specification.

Other downregulated genes in the *Snip1*<sup>Nes</sup>-KO NPCs were involved in the control of self-renewal (Fig. 2i). Characterization of the *Snip1*<sup>Nes</sup>-KO NPCs in vitro showed that, compared with control NPCs, cultured *Snip1*<sup>Nes</sup>-KO NPCs had reduced SOX2 expression (Supplementary Fig. 6a, b). By allowing NPCs to form neurospheres in suspension and through serial passaging, we observed that neurosphere number and cross-sectional area were significantly lower in *Snip1*<sup>Nes</sup>-KO compared with control (Supplementary Fig. 6c–e). Overexpressing human SNIP1

(85% identity with mouse SNIP1) was sufficient to rescue self-renewal in cultured *Snip1*<sup>Nes</sup>-KO NPCs (Supplementary Fig. 6f, g), suggesting functional conservation of SNIP1 between humans and mice. Overall, we conclude that SNIP1 promotes cell survival to maintain NPC self-renewal.

## SNIP1 suppresses differentiation in NPCs

Because at E13.5, the embryonic brain undergoes neurogenesis, we examined the immature neuron marker TUJ1. The relative thickness of the TUJ1-positive region did not significantly differ between *Snip1*<sup>Nes</sup>-KO and control at E13.5 (Supplementary Fig. 7a, b). Considering that

**Fig. 1 | Depletion of SNIP1 in NPCs causes brain dysplasia in mouse embryos.**
**a** WB of control and *Snip1^Nes*-KO NPCs at E13.5. At least 5 replicates of control and *Snip1^Nes*-KO NPCs showed similar results. SNIP1-Ab1; anti-SNIP1 antibody from ProteinTech. **b** IF of MAP2 in control and *Snip1^Nes*-KO embryos at E15. Embryos were cleared by the iDISCO method and imaged with light sheet microscopy. Two replicates of IF showed similar results. Bar, 2 mm. **c** Penetrance of brain dysplasia in E13.5 embryos. Brain dysplasia was determined by the thinning of the brain tissue. Statistical significance was calculated by Fisher's exact test. **d−e** IF of SOX2 and cleaved caspase 3 (CC3) in sagittal cryosections of the E13.5 brain. Germinal zones around lateral ventricle (Lv, forebrain), third ventricle, (midbrain) and fourth (hindbrain) ventricle were examined. Bar, 500 μm. **f**, **g** Quantification of CC3-positive and SOX2-positive cells in the neuroepithelial lining of the ventricles of control and *Snip1^Nes*-KO embryos at E13.5. DAPI staining was used to count the total

number of cells. Each data point represents one image. Eight control embryos and 7 *Snip1^Nes*-KO embryos were analyzed. In (**f**), for lateral ventricle, *n* = 38 images (control) and *n* = 34 (*Snip1^Nes*-KO); for the third ventricle, *n* = 38 (control) and *n* = 36 (*Snip1^Nes*-KO). In (**g**), for lateral ventricle, *n* = 33 (control) and *n* = 35 (*Snip1^Nes*-KO); for the third ventricle, *n* = 32 (control) and *n* = 37 (*Snip1^Nes*-KO). Data are presented as mean ± SEM, and two-way ANOVA was used for statistical analysis. **h−j** IF of SOX2 and CC3 overlayed with neural lineage markers TBR2 and INSM1 of the E13.5 brain. Bar, 50 μm. **i**, **k** Quantification of TBR2-positive or INSM1-positive cells in the neuroepithelial lining of lateral or third ventricles. Each data point represents one image. Five to 8 control embryos and 3-7 *Snip1^Nes*-KO embryos were analyzed. In (**i**), *n* = 43 images (control) and *n* = 30 (*Snip1^Nes*-KO); in (**k**), *n* = 27 (control) and *n* = 17 (*Snip1^Nes*-KO). Data are presented as mean ± SEM, and two-way ANOVA was used for statistical analysis. Source data are provided in a Source Data file (**a**, **f**, **g**, **i**, **k**).

*Snip1^Nes*-KO NPCs were progressively depleted, this lack of difference was a surprise. Therefore, we examined the molecular effect of SNIP1 in differentiating neural cells, which have lost the expression of NPC marker SOX2. We performed RNA-seq of SOX2:GFP-negative cells sorted from E13.5 *Snip1^Nes*-KO and sibling control brains (Fig. 2a, Supplementary Fig. 4a). Using the criteria of fold-change >2 and *p* < 0.05 to compare 2-replicate datasets each from control and *Snip1^Nes*-KO, we identified 658 upregulated genes and 150 downregulated genes in *Snip1^Nes*-KO (Supplementary Fig. 7c). GSEA revealed that upregulated genes in *Snip1^Nes*-KO cells were enriched in functions related to apoptotic clearance, neuronal specification and differentiation, midbrain markers, and known high-CpG-density promoters occupied by bivalent marks (H3K27me3 and H3K4me3) in NPCs[50] (Supplementary Fig. 7d). Downregulated genes in *Snip1^Nes*-KO cells were enriched in functions related to spliceosome, translation and ribosome, nucleosome organization, and apoptosis via p21 but not p53 (Supplementary Fig. 7e). These results suggest that SNIP1 suppresses apoptosis, neuronal specification and differentiation, midbrain genetic programs, and H3K27me3-occupied genes. At E13.5, although upregulated apoptosis reduces *Snip1^Nes*-KO NPCs and intermediate progenitors, the remnant of which give rise to cells that had upregulated neuronal specification and differentiation. These in combination likely lead to no apparent difference in TUJ1-positive cortical thickness between *Snip1^Nes*-KO and control.

Next, we assayed SNIP1 via in vitro differentiation of NPCs. We depleted SNIP1 on either Day 1 or Day 5 by transducing *Snip1*[flox/flox] NPCs with lentiviral control or Cre and profiled gene expression on Day 14 (Supplementary Fig. 7f). Quantitative qPCR showed that compared to control cells, SNIP1-depleted cells upregulated neuronal and glial markers but not NPC markers (Supplementary Fig. 7g, h). The timing of SNIP1 depletion (Day 1 vs. Day 5) did not affect this gene expression pattern (Supplementary Fig. 7g, h). These results further support that SNIP1 suppresses neurogenesis in NPCs. As global knockout (KO) of Snip1 in zebrafish embryos causes reduction in GABAergic and glutamatergic neurons[35], we asked whether SNIP1 depletion alters subneuronal lineage specification in mice. Because of drastic brain tissue loss in *Snip1^Nes*-KO, we could not robustly analyze brain development beyond E13.5. At E13.5, transcript levels of GABAergic neuronal markers *Gad1* and *Slc6a1* were lower in *Snip1^Nes*-KO, and glutamatergic neuronal markers did not differ between *Snip1^Nes*-KO and control (Supplementary Fig. 7i, j). Quantification of immunofluorescence showed significantly lower GABA- (GABAergic neuronal marker) positive cells in *Snip1^Nes*-KO (Supplementary Fig. 7k, l), suggesting that the involvement of SNIP1 in specifying subneuronal lineages may be conserved in higher-order species.

**SNIP1 directly regulates genes with H3K27 modifications**
To determine whether SNIP1 proteins directly bind gene loci to regulate their expression, we profiled the genome-wide distribution of SNIP1 by CUT&RUN[51] in *Snip1^Nes*-KO and control NPCs. Using SICER[52] and MACS2[53] with FDR < 0.05 to compare 2 datasets each from control

and *Snip1^Nes*-KO, we identified 23,188 SNIP1-bound regions in control NPCs and only 4187 regions in *Snip1^Nes*-KO NPCs (Supplementary Fig. 8a−c). The 4187 regions are 18% of those in control and a consequence of remnant SNIP1 proteins in *Snip1^Nes*-KO NPCs (Fig. 1a). Heatmaps showed drastic reduction of SNIP1 CUT&RUN signals in *Snip1^Nes*-KO, suggesting the high specificity of SNIP1 CUT&RUN in NPCs (Fig. 3a). Approximately 50% of SNIP1-bound peaks were within promoters (within 2 kb of transcription start sites), 7.4% were located in exons, 23.3% in introns, 0.7% in transcription termination sites, 9.7% in 5′ distal (2–50 kb from a gene) regions, 3.4% in 3′ distal (2–50 kb from a gene) regions, and 5.5% in intergenic (beyond 50 kb from a gene) regions (Supplementary Fig. 8d). Only 18.6% of SNIP1-bound peaks were located distal to a gene (Supplementary Fig. 8d).

Next, we examined whether SNIP1 occupancy correlates with gene expression. Of the 1210 upregulated genes and 1621 downregulated genes in *Snip1^Nes*-KO, 747 (62%) and 1,271 (78%) were SNIP1 targets, respectively, in control NPCs (Fig. 3b). These overlaps are significant, with *p* = 5.55e-25 and 2.24e-155, respectively, by the hypergeometric test (given a total of 21,636 expressed genes), suggesting that SNIP1 occupies these genes to regulate their expression. GSEA showed that SNIP1 targets that became upregulated in *Snip1^Nes*-KO were enriched in p53 pathway, medulla/midbrain, and apoptosis (Fig. 3c), whereas SNIP1 targets that became downregulated in *Snip1^Nes*-KO were enriched in G2/M checkpoint, cortical development, and chromosome segregation (Fig. 3d). Of the 1,621 downregulated genes in *Snip1^Nes*-KO, 1,093 were SNIP1-bound (not PRC2-bound) and enriched in genetic programs in G2-M checkpoint, mitotic spindle, key signaling pathways for neurodevelopment, and apoptosis (Supplementary Fig. 8e), suggesting that SNIP1 promotes their expression. Intrinsic apoptosis genes were enriched in SNIP1 targets that became upregulated in *Snip1^Nes*-KO (Fig. 3e, f, Supplementary Fig. 8f, g). Of the 44 genes in the intrinsic apoptosis gene set, 37 promoters were bound by SNIP1 in control NPCs (*p* = 4.62e-5; Fig. 3g), suggesting that SNIP1 directly suppresses these genes. These data suggest that SNIP1 directly regulates genetic programs crucial to apoptosis and cell cycle.

Upregulated genes in *Snip1^Nes*-KO were enriched in genes whose high-CpG-density promoters that are 1) H3K27me3-occupied in the embryonic murine brain[50] or 2) bivalent in mouse NPCs[54] (Supplementary Fig. 9a, b). This prompted us to scrutinize whether SNIP1 controls genetic programs through H3K27 modifications. We profiled H3K27me3 and H3K27ac by CUT&RUN in *Snip1^Nes*-KO and control SOX2-positive NPCs (Supplementary Fig. 9c−g). We observed a strong correlation between upregulated genes, lower H3K27me3 occupancy, and higher H3K27ac occupancy, whereas downregulated genes had higher H3K27me3 occupancy and lower H3K27ac occupancy in *Snip1^Nes*-KO (Fig. 3c, d, Supplementary Fig. 9h, i). Among the 44 upregulated intrinsic apoptosis genes, 9 genes showed reduced H3K27me3 occupancy, and 5 genes showed increased in H3K27ac occupancy (*p* < 0.05, Fig. 3g, h, Supplementary Fig. 9j, k). These data suggest that SNIP1 at chromatin controls H3K27 modifications and gene expression.

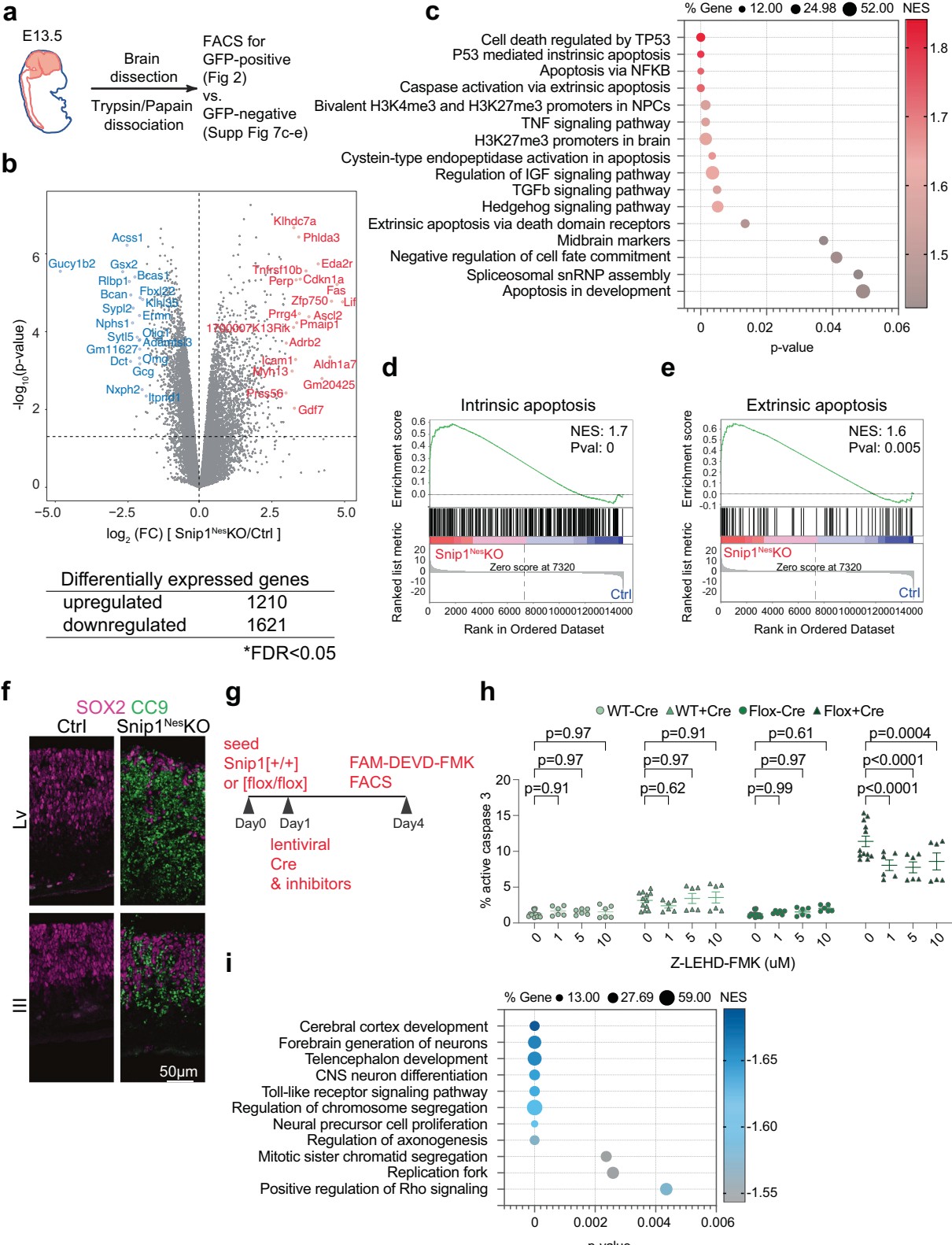

## TGFβ and NFκB signaling pathways control SNIP1 localization to chromatin

As SNIP1 participates in TGFβ[22–25] and NFκB[26–30] signaling pathways, we aimed to test whether their inhibition affects apoptosis in SNIP1-depleted NPCs. We tested 3 inhibitors to TGFβ or NFκB signaling for toxicity to NPCs (Fig. 4a, b). At 0.1–0.5 μM concentrations, K02288 targeting TGFβ (but not Galunisertib or LDN-193189) consistently

reduced apoptosis in SNIP1-depleted NPCs (Fig. 4c–e). In contrast, 2 inhibitors to TGFβ signaling and 3 inhibitors to NFκB signaling increased apoptosis in NPCs (Fig. 4f–h). Therefore, we decided to test whether any of the 6 inhibitors alters SNIP1 binding to chromatin. We treated NPCs with DMSO or inhibitors at 0.5 μM for 3 days and performed CUT&RUN to assay SNIP1 localization on chromatin (Fig. 4i). Analyzing 2 replicate SNIP1 CUT&RUN data per treatment condition,

**Fig. 2 | SNIP1 suppresses genes involved in apoptosis and signal transduction and promotes genes for brain development. a** Schematic of the brain NPC collection. **b** Volcano plot and the number of differentially expressed genes between the control and *Snip1[Nes]*-KO NPCs. The table shows the number of genes that passed the cutoff of FDR < 0.05. FDR was calculated by the Benjamini & Hochberg method. *P* values were calculated by two-sided Voom-limma *t* test. **c, i** Bubble plots of the enriched gene sets in upregulated genes and downregulated genes in *Snip1[Nes]*-KO vs. control NPCs. Differentially expressed genes were first ranked by their fold-change, *p*-value, and expression level before Gene Set Enrichment Analysis (GSEA) was performed. *P*-values were calculated by a right-sided permutation test with FDR adjustment. **d, e** Representative GSEA of upregulated genes in *Snip1[Nes]*-KO vs. control NPCs. Upregulated genes were enriched in gene sets related to both intrinsic and extrinsic apoptosis. Differentially expressed genes were first ranked by

their fold-change and *p*-value before GSEA was performed. **f** IF of cleaved caspase 9 (CC9) overlayed with SOX2 in sagittal cryosections of the E13.5 brain. Bar, 50 μm. **g** Schematic of transduction with mCherry-Cre lentivirus and treatment with inhibitors in *Snip1*[+/+] and *Snip1*[flox/flox] NPCs. **h** The percentage of cells with active caspase 3 quantified by FACS. Caspase 9 inhibitor (Z-LEHD-FMK) was added at different concentrations along with mCherry-Cre lentivirus. The percentage of FAM-FLICA (active caspase 3)-positive population (out of total population) is shown. *N* = 12 for DMSO control and *n* = 6 for the rest of the sample. Data are presented as mean ± SEM, and two-way ANOVA was used for statistical analysis. The gating strategy and representative FACS plots are shown in Supplementary Information (Supplementary Fig. 4e, Supplementary Fig 14a). Source data are provided in a Source Data file (**c, h, i**).

we used *p* < 0.05 to identify significant and consistent differences in SNIP1 CUT&RUN signals between inhibitor and control treatments. We identified SNIP1 CUT&RUN changes induced by K02288 treatment (Supplementary Data 1). We also identified significant changes in SNIP1 CUT&RUN at SNIP1-bound genes that overlap in any 2 inhibitor treatments (Supplementary Data 1). Average profiling of SNIP1 CUT&RUN signals at SNIP1-bound promoters in NPCs treated with different inhibitors confirmed that inhibition to TGFβ (Fig. 4j, k) and NFκB (Fig. 4l, m) signaling significantly altered SNIP1 binding to promoters. These data suggest that TGFβ and NFκB signaling pathways control SNIP1 binding to specific gene loci in NPCs.

## PRC2 requires SNIP1 for localization to chromatin and H3K27me3 deposition

We investigated the interactions between SNIP1, H3K27 methyltransferase PRC2, and histone acetyltransferases p300 and CBP. Anti-SNIP1 antibody co-immunoprecipitated with known PRC2 subunits Jumonji and AT-rich interaction domain containing 2 (JARID2), Suppressor of zeste 12 (SUZ12), and EZH2, but not the negative control RBBP5 in the NPC nuclear extract (Fig. 5a). Anti-SNIP1 antibody did not co-immunoprecipitate with PRC2 subunits in SNIP1-depleted NPCs (Supplementary Fig. 10a), supporting specificity of the antibody. Anti-JARID2, EZH2, or EED antibody co-immunoprecipitated SNIP1 and other PRC2 subunits but not the negative control RBBP5 in the NPC nuclear extract (Fig. 5b–d). Anti-p300 or CBP antibody failed to co-immunoprecipitate SNIP1, suggesting that in NPCs, their physical interaction is undetectable (Supplementary Fig. 10b, c). The PRC2–SNIP1 binding is supported by other proteomics studies that showed PRC2 co-immunoprecipitates SNIP1[19,20].

We tested whether SNIP1 alters the expression of PRC2 subunits. Although SNIP1 depletion lowered *Ezh2* transcript levels, it did not alter the protein levels of PRC2 subunits, H3K27me3, or H3K27ac (Supplementary Fig. 10d–g). Next, we performed CUT&RUN to profile SUZ12 and EZH2 in *Snip1[Nes]*-KO and control NPCs (Supplementary Fig. 10h–j). Consistent with the co-immunoprecipitation results, SUZ12, and EZH2 co-occupy SNIP1-bound target sites genome-wide (Fig. 5e). To analyze PRC2–SNIP1 interactions on chromatin, we performed CUT&RUN-reChIP. In this assay, chromatin released by SNIP1 CUT&RUN were immunoprecipitated by IgG, EZH2, or H3K27me3. The representative loci *Mcm7*, *Aen*, *Lhx8*, and *Eomes* had co-occupancy of SNIP1 with EZH2 and H3K27me3 but not negative control IgG (Fig. 5f). Genome wide, CUT&RUN-reChIP showed that EZH2 and H3K27me3 had high overlaps with SNIP1 and PRC2 at SNIP1-bound peaks (Fig. 5e).

In examining the role of SNIP1 in PRC2 chromatin occupancy, we found that the levels of SUZ12, EZH2, and H3K27me3 at chromatin were significantly reduced in *Snip1[Nes]*-KO (Fig. 5g, Supplementary Figs. 10k, 11a). In contrast, H3K27ac levels were less altered in *Snip1[Nes]*-KO NPCs (Fig. 5g, Supplementary Figs. 10k, 11a). Next, we used an in vitro assay to analyze the kinetic effect of SNIP1 depletion on PRC2. Compared with control lentivirus, lentiviral Cre transduction of *Snip1*[flox/flox] NPCs depleted SNIP1 transcripts by 70% and 99.9% on

the second and third day, respectively. This did not alter the transcript level of PRC2 components (Supplementary Fig. 11b). Using EZH2 and SUZ12 CUT&RUN, we observed a strong reduction of PRC2 on chromatin by the third day of SNIP1 depletion (Supplementary Fig. 11c). Together, these data support that PRC2 requires SNIP1 for binding to chromatin.

To characterize the SNIP1-PRC2 targets, we performed de novo motif discovery by the HOMER software[55]. Using the criteria of fold-change >2 and *p* < 0.05 in control vs *Snip1[Nes]*-KO, SNIP1 targets were enriched in motifs of E2F proteins, SP1/3/4, and EGR1 (Fig. 5h). Motifs of previously reported SNIP1 interactors, SMAD proteins and RELA, were also found amongst SNIP1 targets[21,26]. In SUZ12-bound peaks that had reduced binding in *Snip1[Nes]*-KO, motifs were enriched with SP2, RELA, E2F proteins, EGR1, HINFP, PLAGL1, and NF-Y subunits (Fig. 5i). The similarities of SNIP1- and SUZ12-bound motifs point to potential interactions of SNIP1 and PRC2 with some of these transcription factors.

We next examined whether the SNIP1- or SUZ12-bound motifs were overrepresented in differentially expressed genes. Using Enrichr[56], we found that upregulated genes in *Snip1[Nes]*-KO were targets of TP53, FLI1, SUZ12, MAX, and MYC, whereas downregulated genes were targets of E2F4, SOX2, NFYB and NFYA (Fig. 5j, k). E2F proteins were uncovered by motif and Enrichr analyses, and E2F4 targets *Mcm7* and *Anp32e* were SNIP1 and PRC2 targets that had reduced binding in *Snip1[Nes]*-KO (Fig. 5k, Supplementary Fig. 11d). Among upregulated genes in *Snip1[Nes]*-KO, MYC and MAX targets were identified by both motif and Enrichr analyses (Fig. 5j, Supplementary Fig. 11e). These data suggest that E2F proteins, MYC, and MAX may influence SNIP1-PRC2 activities for gene regulation.

## PRC2 promotes apoptosis in the absence of SNIP1

We tested the role of PRC2 in NPC survival in vivo. We used *Nes*::Cre to excise exons 3 to 6 of *Eed* (a PRC2 core subunit) to generate *Eed[Nes]*-KO and *Snip1[Nes]*-*Eed[Nes]*-dKO (Supplementary Fig. 12a, b). EED depletion in NPCs did not induce apoptosis in *Eed[Nes]*-KO E13.5 brain (Supplementary Fig. 12c–g). Others had studied EED in the control and *Eed[Emx1]*-KO dorsal telencephalon and made conclusions[57] that differed from our observations in *Eed[Nes]*-KO E13.5 brain. This difference could be explained by the expression of *Emx1*::Cre and *Nes*::Cre at different developmental stages and brain regions. Apoptosis was observed in other brain regions of control and *Eed[Nes]*-KO brains (Supplementary Fig. 12c, d). As EED has already been shown to affect neurogenesis at E14.5 but not E16.5[57], the exact effect of EED/PRC2 on NPC functions is developmental stage- and cell context-dependent. Our analysis showed that compared with *Snip1[Nes]*-KO, *Snip1[Nes]*-*Eed[Nes]*-dKO had significantly fewer cl-caspase 3-positive cells (Fig. 6a–c), more SOX2-positive NPCs (Fig. 6d, e), and more TBR2- or INSM1-positive intermediate progenitors (Fig. 6f–i). TUJ1-positive immature neurons were not markedly affected by the SNIP1–PRC2 functional interaction (Supplementary Fig. 12h, i). EED depletion in the *Snip1[Nes]*-KO embryonic brain reduced apoptosis and

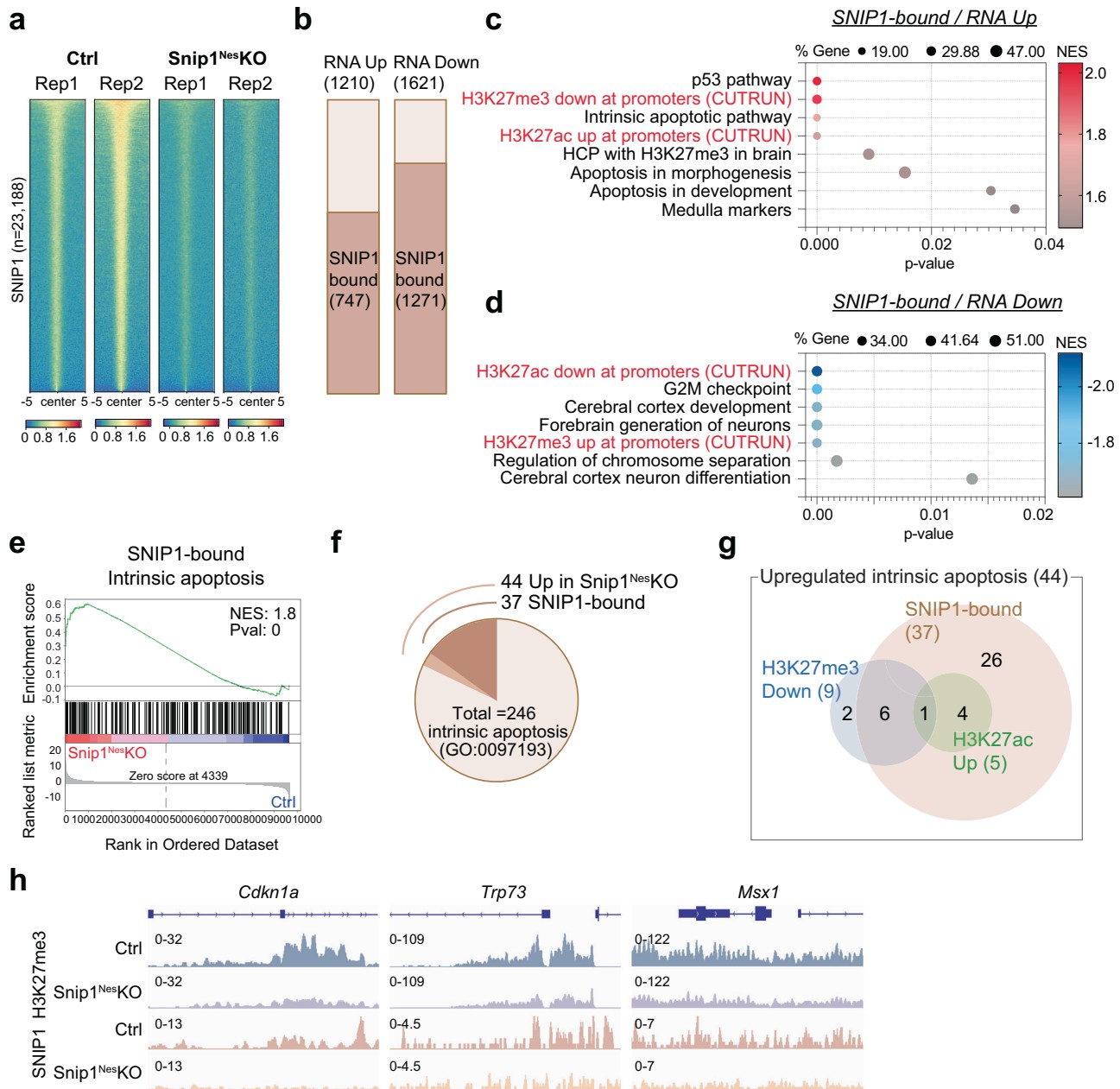

**Fig. 3 | SNIP1 binds to chromatin and affects H3K27me3 levels for regulating gene expression. a** Heatmaps representing the binding intensity of 2 biological replicates of SNIP1 CUT&RUN in control and *Snip1^Nes*-KO NPCs. Binding intensity for 5 kb on either side of all 23,188 SNIP1 CUT&RUN peaks are shown. Blue indicates low intensity and red indicates high intensity. **b** Bar charts displaying the numbers of upregulated and downregulated genes (*Snip1^Nes*-KO vs. control, FDR < 0.05) that are bound by SNIP1 at their gene body. **c, d** Bubble plots of the enriched gene sets in SNIP1-bound genes that became upregulated genes and downregulated genes in *Snip1^Nes*-KO vs. control NPCs. When adding our H3K27me3/ac CUT&RUN data to the GSEA gene sets, H3K27me3 and H3K27ac levels showed anti-correlation and correlation with gene expression, respectively. Source data are provided in a Source Data file. **e** Representative GSEA of upregulated genes in *Snip1^Nes*-KO NPCs vs. control NPCs that are bound by SNIP1 in control NPCs. Intrinsic apoptosis genes

were mostly SNIP1-bound and were enriched in the upregulated genes in *Snip1^Nes*-KO NPCs. Differentially expressed genes were first ranked by their fold-change and *p*-value before GSEA was performed. **f** Pie chart showing the proportions of intrinsic apoptosis genes that are upregulated in *Snip1^Nes*-KO NPCs and/or bound by SNIP1. **g** Venn diagram displaying the numbers of upregulated intrinsic apoptosis genes in 3 categories. Using our SNIP1, H3K27me3, and H3K27ac CUT&RUN data, the 44 genes were categorized into 1) SNIP1-bound in control NPCs, 2) reduced H3K27me3 levels in *Snip1^Nes*-KO vs. control NPCs (*p* < 0.05), and/or 3) increased H3K27ac levels in *Snip1^Nes*-KO vs. control NPCs (*p* < 0.05). **h** H3K27me3 and SNIP1 CUT&RUN tracks visualized by Integrative Genomics Viewer (IGV) at upregulated intrinsic apoptosis genes. *Cdkn1a*, Chr17: 29,090,888 − 29,095,850. *Trp73*, Chr4: 154,132,565 − 154,143,373. *Msx1*, Chr5: 37,818,429 − 37,828,924.

rescued NPCs and intermediate progenitors. Therefore, SNIP1 suppresses apoptosis in the developing brain by counteracting PRC2.

We profiled the transcriptomes of control, *Snip1^Nes*-KO, and *Snip1^Nes*-*Eed^Nes*-dKO brain tissues. Unsupervised clustering based on top 3,000 most differentially expressed genes (based on median variation

values) suggests a higher similarity in the transcriptomes between control and *Snip1^Nes*-*Eed^Nes*-dKO compared with *Snip1^Nes*-KO (Fig. 6j). Using fold-change >2 and *p* < 0.05 to compare datasets from *Snip1^Nes*-KO and *Snip1^Nes*-*Eed^Nes*-dKO, we identified 184 upregulated genes and 994 downregulated genes in *Snip1^Nes*-*Eed^Nes*-dKO (Supplementary Fig. 13a).

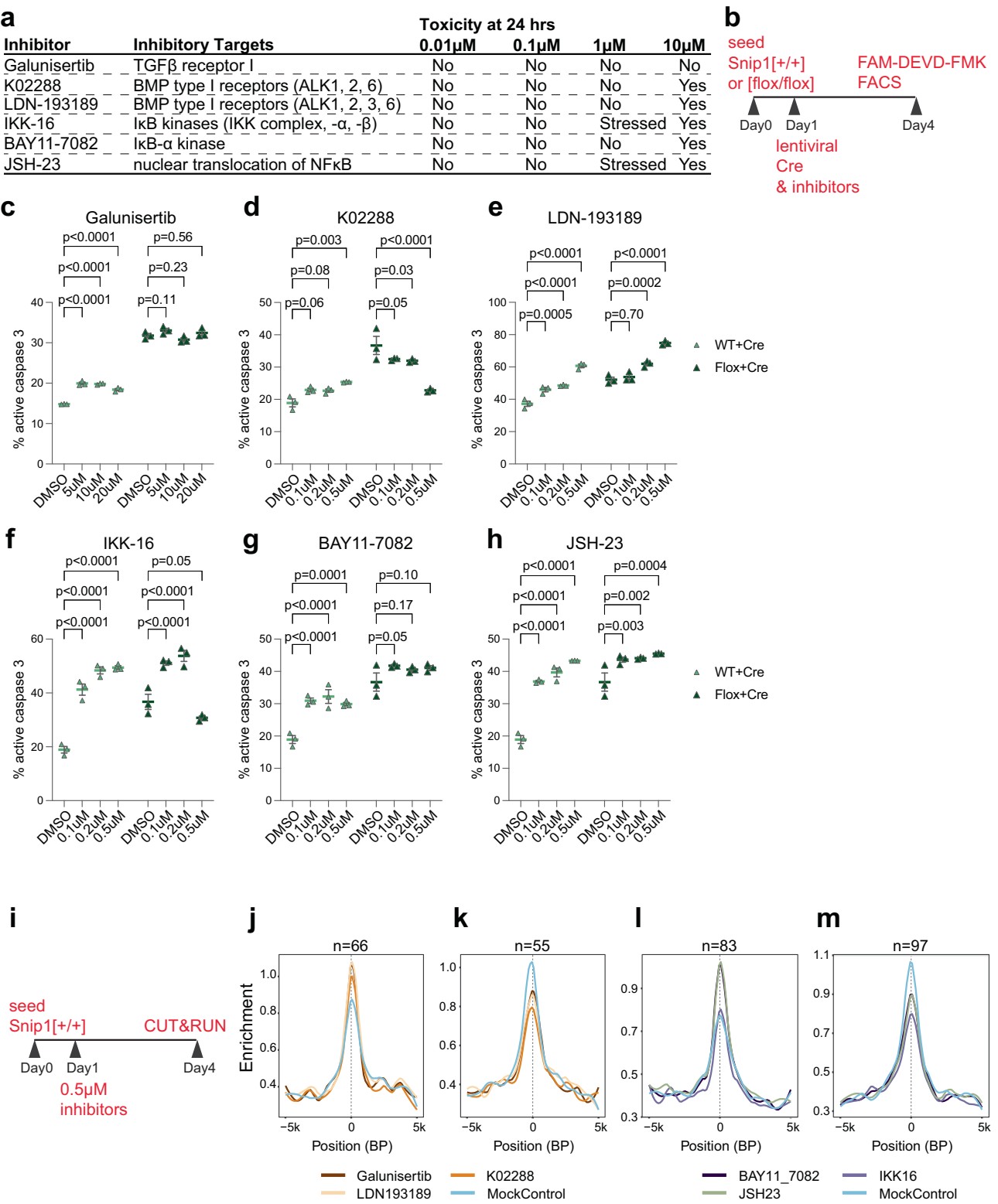

To provide a potential molecular explanation of the rescue of NPCs and intermediate progenitors in *Snip1^Nes*-*Eed^Nes*-dKO, we analyzed differentially expressed genes among control, *Snip1^Nes*-*Eed^Nes*-dKO, and *Snip1^Nes*-*KO*. We found that 197 downregulated genes in *Snip1^Nes*-*KO* partially regained expression in *Snip1^Nes*-*Eed^Nes*-dKO (Fig. 6k). These rescued genes are involved in G2/M checkpoint, E2F targets, cholesterol homeostasis, mTORC1 signaling, and androgen response (Fig. 6l). In contrast, 32 upregulated genes in *Snip1^Nes*-*KO* became downregulated in *Snip1^Nes*-*Eed^Nes*-*KO* (Fig. 6m) and are involved in p53

pathway, inflammatory and interferon gamma responses, NFκB signaling, and apoptosis (Fig. 6n). Of these 32 genes, *Cdkn1a* is the only intrinsic apoptosis gene with significantly lower H3K27me3 levels in *Snip1^Nes*-*KO* versus control (FDR < 0.05; Supplementary Fig. 13b), suggesting that SNIP1 directly promotes PRC2 and H3K27me3 at the *Cdkn1a* locus. Our profiling of SNIP1-PRC2 genome occupancy suggests that SNIP1 inhibits H3K27me3 deposition at loci that include E2F targets (Supplementary Fig. 11d) but promotes H3K27me3 deposition at other loci that include apoptotic genes *Cdkn1a*, Nkx2-9, *Etv4*, *Pxdc1*,

**Fig. 4 | Inhibitors to TGFβ and NFκB signaling pathways alter NPC survival and SNIP1 binding to chromatin. a** Inhibitors targeting components in TGFβ and NFκB signaling pathways and their cytotoxicity at different concentrations. **b** Schematic of inhibitor assay. On Day 1, WT or *Snip1*[flox/flox] NPCs were treated with inhibitor or DMSO and transduced with lentiviral Cre for SNIP1 depletion. FAM-DEVD-FMK was used for assaying cl-caspase 3 by FACS on Day 4. **c–h** The percentage of cells with active caspase 3 quantified by FACS. Inhibitors were added at different concentrations along with mCherry-Cre lentivirus. The percentage of FAM-FLICA (active caspase 3)-/ mCherry-double positive population (out of mCherry-positive population) is shown. *N* = 3 for each treatment. Data are presented as mean ± SEM,

and two-way ANOVA was used for statistical analysis. The representative FACS plots are shown in Supplementary Fig. 14c, d. Source data are provided in a Source Data file. **i** Schematic of SNIP1 CUT&RUN with inhibitor treatment. At day 1, NPCs were treated with DMSO control or different inhibitors. Replicate SNIP1 CUT&RUN was performed for each of the 7 treatments on day 4. **j–m** Profile plots comparing the median binding intensity of SNIP1 in NPCs at the SNIP1-bound targets that had significantly higher or lower SNIP1 binding in inhibitors versus DMSO control treatment. *n* indicates region numbers. Regions were considered true SNIP1 targets when SNIP1 levels reduced in *Snip1^Nes*-KO vs. control NPCs with *p* < 0.05.

and *Tap1* (Supplementary Fig. 13b, c). Taken together, SNIP1 exerts loci-dependent control of H3K27me3 deposition and gene expression to balance of NPC division, apoptosis, and differentiation in the developing brain.

## Discussion

Using embryonic mouse brains as a model, we have uncovered that SNIP1 promotes neurogenesis and suppresses intrinsic apoptosis. SNIP1 epigenetically regulates these processes by binding to chromatin, which was partially guided by TGFβ and NFκB signaling pathways. Our study further shows SNIP1 and PRC2 co-occupy chromatin targets to regulate genetic programs. Initially, we had hypothesized that SNIP1 directly suppresses apoptotic genes in a cooperative manner with PRC2. However, the rescue of *Snip1^Nes*-KO NPCs by EED/PRC2 depletion came as an intriguing surprise and suggest that SNIP1 and PRC2 have a balancing relationship for regulating genetic programs in the brain. Indeed, some of the rescued genes by *Snip1^Nes*-*Eed^Nes*-dKO versus *Snip1^Nes*-KO have high relevance. CDKN1B/p27 and CDKN1A/p21 regulate cell cycle progression[58–61] and apoptosis[62–65]. CDC25B[66–68], CDK4[69–71], and PTPN14[72] are also key to cell cycle progression. HDAC2 regulates transcription and brain development[73,74], and *SMS* mutations are causally linked to neurodevelopmental defects in Snyder-Robinson Syndrome[75–77]. The PRC2–SNIP1 interaction regulates these and other potentially crucial genes in cell cycle progression, apoptosis, or brain development.

A role of SNIP1 in anti-apoptosis has been implicated by other studies. SNIP1 depletion induced defective splicing and apoptosis in zebrafish embryos[35]. SNIP1 depletion in a human osteosarcoma cell line U2OS increases sensitivity to cisplatin-induced apoptosis[78]. Our study adds to the SNIP1 mechanism: (1) the participation of caspases 9 and 3, (2) the chromatin-based role of SNIP1, (3) TGFβ and NFκB signaling influences the genomic distribution of SNIP1, (4) the balancing relationship between SNIP1 and PRC2 to fine-tune genetic programs, (5) implicating the participation of E2F proteins, SMADs, and RELA/B, and (6) the requirement of SNIP1 for PRC2 binding to chromatin and H3K27me3 deposition in NPCs. Nevertheless, the precise mechanisms by which SNIP1–PRC2 localizes to chromatin targets and how the downstream factors orchestrate caspase 9-dependent apoptosis remain unclear and subject to extensive future studies.

The role of caspase 9 and intrinsic apoptosis in brain development has been little understood, despite the human genetic evidence connecting loss of function in caspase 9 to neural tube defects[79–81] and pediatric brain tumors[82,83]. Global KO of either caspase 3 or 9 in mice leads to prenatal death with brain malformation, including neural tube closure defects and exencephaly[84–86]. These findings point to a conserved requirement of caspase 3 and 9 for brain development. Our study implicates the essential roles of SNIP1 and PRC2 in dampening the activities of caspase 9 and caspase 3 in order to balance the division and death of NPCs. And this dampening is potentially mediated through p53, whose protein levels markedly increase and coincide with higher expression of p53 targets in *Snip1^Nes*-KO NPCs.

Our study identifies a physiological role of SNIP1 in regulating cell cycle that was previously implicated by cell line studies[31,87,88]. We showed that *Snip1^Nes*-KO NPCs have defective genetic programs in

chromosome segregation, cell proliferation, and replication fork. In culture, *Snip1^Nes*-KO NPCs diminished the growth of neurospheres over time. We previously showed that PRC2 maintains the self-renewal of NPCs by suppressing neurogenesis[18]. This study uncovered that SNIP1 and PRC2 transcriptionally regulate cell survival; this may be clinically relevant to developmental defects including skull dysplasia, global developmental delay, and intellectual disability and seizure that are associated with 1097 A > G (Glu366Gly) variant of *SNIP1*[36,37]. Beyond embryogenesis, SNIP1–PRC2 may regulate other intrinsic properties of NPCs and neurons. One example is cell type-specific control of synapse formation and neuronal wiring, which is essential to neural circuitry and function[89]. Current and future findings about SNIP1–PRC2 interactions in NPC survival and maturing neurons will improve our understanding about neurodevelopmental disorders caused by *SNIP1* mutations and PRC2 dysfunction.

## Methods

### Buffers
PBS: 137 mM NaCl, 2.7 mM KCl, 10 mM phosphate buffer (pH 7.4)
PBST: PBS with 0.1% Triton X-100
HEPM (pH 6.9): 25 mM HEPES, 10 mM EGTA, 60 mM PIPES, 2 mM MgCl$_2$
IF blocking buffer: 1/3 Blocker Casein (ThermoFisher 37528), 2/3 HEPM with 0.05% Triton X-100
Buffer A: 10 mM HEPES (pH 7.9), 10 mM KCl, 1.5 mM MgCl$_2$, 0.34 M sucrose, 10% glycerol
Buffer D: 400 mM KCl, 20 mM HEPES, 0.4 mM EDTA, 20% glycerol
iDISCO PTx.2: PBS with 0.2% Triton X-100
iDISCO PTwH: PBS with 0.2% Tween-20 and 10 µg/mL heparin
iDISCO Permeabilization solution: PTx.2 with 306 mM glycine and 20% DMSO
iDISCO Blocking solution: PTx.2 with 6% donkey serum and 10% DMSO
CUT&RUN Binding buffer: 20 mM HEPES-KOH (pH 7.9), 10 mM KCl, 1 mM CaCl$_2$, 1 mM MnCl$_2$
CUT&RUN Wash buffer: 20 mM HEPES (pH 7.5), 150 mM NaCl, 0.5 mM spermidine, Protease Inhibitor Cocktail (Sigma-Aldrich 11873580001)
CUT&RUN Digitonin block buffer: CUT&RUN Wash buffer with 2 mM EDTA and 0.05% digitonin
CUT&RUN 2X Stop buffer stock: 340 mM NaCl, 20 mM EDTA, 4 mM EGTA, 0.02% digitonin
CUT&RUN Stop buffer: Into 1 mL of 2X Stop buffer stock, add 5 µL of 10 mg/mL RNase A and 133 µL of 15 mg/mL GlycoBlue™ Coprecipitant (ThermoFisher AM9516)
ChIP elution buffer: 50 mM Tris-Cl (pH 8), 10 mM EDTA, 1% SDS
Antibodies used in this study are listed in Supplementary Table 1.

### Animals
All animal experiments were approved by the Institutional Animal Care and Use Committee at St. Jude Children's Research Hospital under protocol 573 and were conducted in accordance with ethical guidelines for animal research. To generate conditional knockout embryos, we used the Cre/lox system. *Snip1*-tm1a (Infrafrontier/EMMA 04224) were first crossed with *Actin*-FLPe to generate the *Snip1*-flox line. To genetically label NPCs, *Snip1*-flox; *Nestin*-Cre mice were crossed with

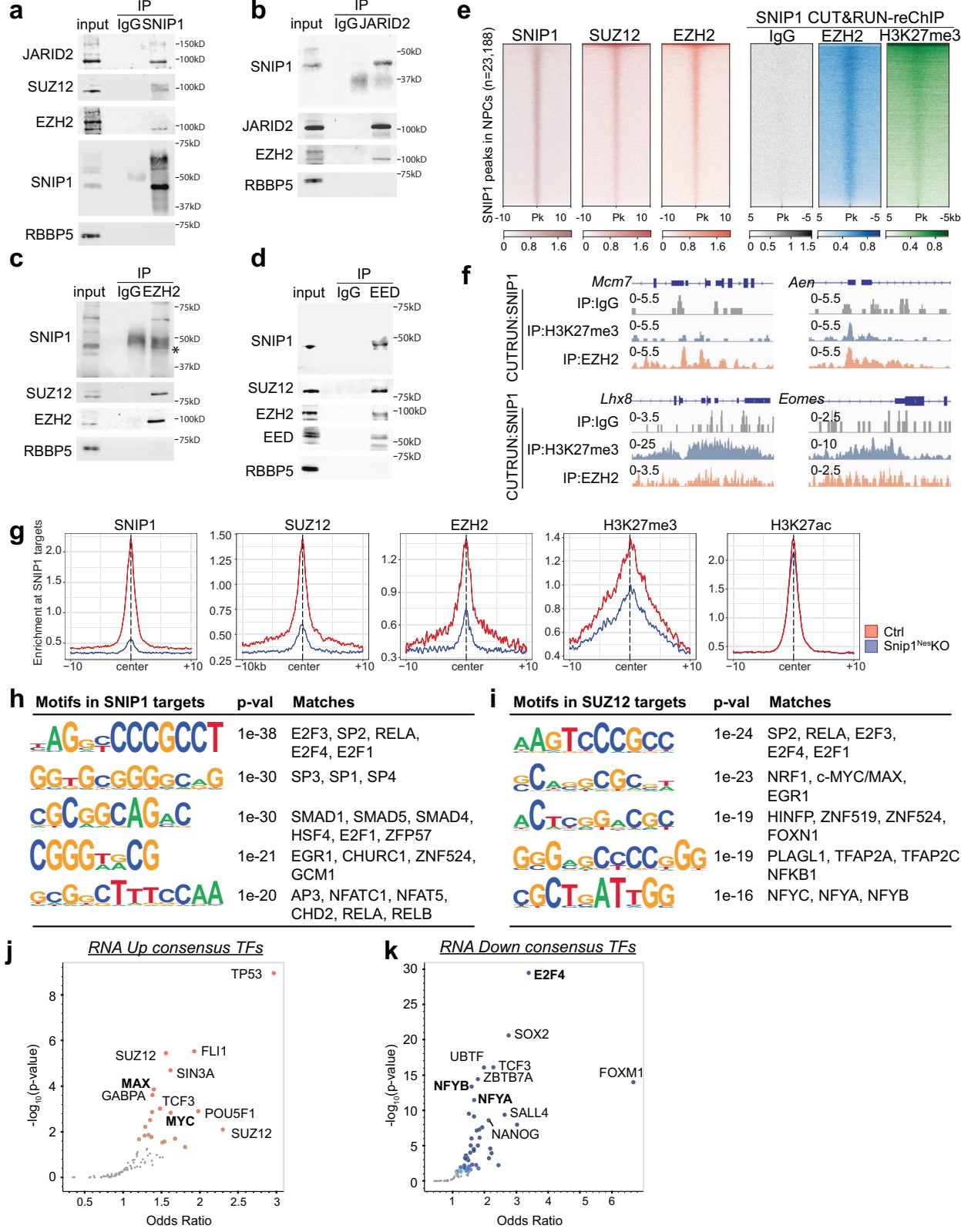

Sox2-eGFP transgenic mice. Animals were housed in an enriched environment at 23 °C, 30–70% humidity, and with a 12 h light /12 h dark cycle. Embryos were harvested at embryonic days as indicated in the figures or text. For all the animal experiments, both sexes were included. The following mouse lines were used in this study and genotyping primers and conditions are shown in Supplementary Table 2.

Snip1-tm1a: B6Dnk;B6N-Snip1<tm1a(EUCOMM)Wtsi >/H (Infra-frontier/EMMA 04224)
Eed-flox: B6;129S1-Eed^tm1Sho/J (JAX Stock 022727)[90]
Actin-FLPe: B6;SJL-Tg(ACTFLPe)9205Dym/J (a gift from Dr. Peter McKinnon at St. Jude Children's Research Hospital, JAX Stock 003800)[91]

**Fig. 5 | SNIP1-bound regions are co-occupied by PRC2 on NPC chromatin. a–d**
Co-immunoprecipitation followed by WB to examine the interaction between SNIP1
and PRC2. (**a**) SNIP1, (**b**) JARID2, (**c**) EZH2, or (**d**) EED was immunoprecipitated in the
NPC nuclear extract. RBBP5 was a negative control. Source data are provided as a
Source Data file. **e** Heatmaps aligning chromatin peaks enriched with SNIP1, SUZ12,
and EZH2 in NPCs. Peaks from SNIP1 CUT&RUN−reChIP with IgG, EZH2, and
H3K27me3 were aligned to SNIP1-bound peaks. Intensity for 5 or 10 kb on either
side of 23,188 SNIP1-bound peaks are shown. A dark color indicates high intensity
and a light color indicates low intensity. **f** Tracks of SNIP1 CUT&RUN−reChIP with
IgG, EZH2, and H3K27me3 are visualized by Integrative Genomics Viewer (IGV).
*Mcm7*, Chr5: 138,169,717 − 138,173,621. *Aen*, Chr7: 78,894,526 − 78,898,271. *Lhx8*,
Chr3: 154,325,066 − 154,334,835. *Eomes*, Chr9: 118,474,178 − 118,480,775. **g** Profile

plots comparing the median binding intensity of SNIP1, PRC2, and H3K27me3/ac in
*Snip1^Nes*-KO vs. control NPCs at the SNIP1 targets. Regions were considered true
SNIP1 targets when SNIP1 levels were reduced in *Snip1^Nes*-KO vs. control NPCs with
$p < 0.05$. **h, i** Motifs of SNIP1- and SUZ12-bound regions where their levels sig-
nificantly reduced in *Snip1^Nes*-KO NPCs with fold-change >2 and $p < 0.05$. HOMER de
novo analysis was performed and the five motifs with lowest $p$-values and had
vertebrate motif matches are listed here. **j, k** Volcano plots of transcription factors
whose binding to our differentially expressed genes (FDR < 0.05) has been repor-
ted. Genes were searched against ENCODE and ChEA consensus TFs from ChIP-X
database using Enrichr[56]. Darker colors show smaller $p$-values and large points
passed $p$-value < 0.05. Transcription factors in bold were found in our CUT&RUN
motif analyses in Fig. 5h, i.

*Nestin*-Cre: B6.Cg-Tg(*Nes*-Cre)1Kln/J (JAX Stock 003771)[92]
*Emx1*-Cre: B6.129S2-Emx1^tm1(cre)Krj/J (a gift from Dr. Peter McKinnon
at St. Jude Children's Research Hospital, JAX Stock 005628)[93]
*Sox2*-eGFP: B6;129S1-*Sox2*^tm1Hoch/J (JAX Stock 017592)[94].

### Isolation and culturing of mouse NPCs

To obtain mouse brain cells, embryos at an indicated embryonic day
were dissected out from the uterus and visceral yolk sac. A part of the
tail or limbs was collected for genotyping. Brains were dissected from
embryos under the dissection microscope in cold 1x PBS. Then, 300 µL
Dulbecco's Modified Eagle's Medium (DMEM) (ATCC 30-2002) and
150 µL of 10 mg/mL collagenase Type II (Worthington LS004176) were
added to each brain and incubated for 5–10 min at 37 °C. After cen-
trifugation at $1000 \times g$ for 3 min, the tissue was incubated with 500 µL
0.25% Trypsin-EDTA (ThermoFisher 25200056) for 5 min at 37 °C.
Trypsinization was quenched with 500 µL DMEM supplemented with
10% fetal bovine serum (FBS) and pelleted by centrifugation at
$1000 \times g$ for 3 min. Alternatively, cells were dissociated from the brain
using the papain dissociation system (Worthington LK003153). Cells
were then resuspended in 500 µL of NPC culture media (NeuroCult™
Proliferation Media; STEMCELL Technologies 05702) supplemented
with 30 ng/mL human recombinant epidermal growth factor (rhEGF)
(STEMCELL Technologies 78006) and filtered through a 40 µm filter
(Fisherbrand™ 22-363-547) to obtain single cells. To collect NPCs, the
dissociated brain cells were cultured in ultra-low attachment 6-well
plates (Corning® Costar® CLS3471) in the NPC culture media at 37 °C.
NPCs formed neurospheres in suspension. For passaging, neuro-
spheres were incubated with Accutase™ (STEMCELL Technologies
07920) for 5 min at 37 °C and then dissociated by pipetting. After
adding an equal volume of the NPC culture media, dissociated cells
were centrifuged at 500 x g for 5 min. Cells were then grown in the NPC
culture media either in suspension in the ultra-low attachment 6-well
plates or on Matrigel (Corning™ 354230) -coated plates. The medium
was changed every 2 to 3 days. For collecting uncultured NPCs, cells
from the *Sox2*-eGFP brains or stained with NeuroFluor™ CDr3 (STEM-
CELL Technologies 01800) were sorted by fluorescence-activated cell
sorting (FACS).

### Neurosphere assay

NPCs were seeded into ultra-low attachment 6-well plates and grown in
the NPC culture media at 37 °C with 250 µL of media added every
2 days. Neurospheres were imaged after 5 days of culturing. For a
neurosphere rescue experiment, freshly dissociated brain cells were
first transduced with lentivirus delivering human SNIP1 transgene
(System Biosciences CD823A-1) in ultra-low attachment 6-well plates.
After 5 days, transduced neurospheres were then dissociated and
seeded for a neurosphere assay. All the neurosphere assays were done
in three replicates. At least 8 images of neurospheres per well were
captured with ZEISS AxioObserver D1 at 5x magnification. The area and
the number of neurospheres were quantified by using FIJI. To generate
clean binary images, images were processed with "Process" -> "Find
edges" followed by "Image" -> "Adjust" -> "Threshold". After inverting

the images, the number and the area of the neurospheres were
obtained by selecting "Analyze" -> "Analyze particles". If multiple
neurospheres were too close for the software to quantify individually,
one of the two methods was applied after generating the binary ima-
ges: manual quantification by drawing the outline of each neurosphere
and selecting "Analyze" -> "Measure", or computationally separating
the neurospheres by selecting "Process" -> "Binary" -> "Fill Holes,"
followed by "Process" -> "Binary" -> "Watershed".

### In vitro neural differentiation assay

For neural differentiation, NeuroCult™ Differentiation Kit (STEMCELL
Technologies 05704) was used by following the manufacturer's
instructions. On Day 0, approximately $1.6 \times 10^6$ NPCs from *Snip1*[flox/
flox] embryos were seeded onto each well of 6-well plates which were
double-coated with 10 µg/mL of poly-D-lysine and 10 µg/mL laminin.
On Day 1 (or Day 5), cells were incubated with either mCherry-control
or mCherry-Cre lentivirus (Vector Core Lab at St. Jude Children's
Research Hospital) for 8 h, washed twice with 1X PBS, and cultured for
14 days. During culturing, a half-medium change was done when the
culture media turned yellow. On Day 14, cells were harvested for RNA
extraction and real-time quantitative PCR.

### Inhibitor treatment and FACS-based cell death assay

$5 \times 10^5$ NPCs from *Snip1*[+/+] and *Snip1*[flox/flox] embryos were seeded
onto each well of matrigel-coated 6-well plates. On the following day,
cells were incubated with mCherry-Cre lentivirus (Vector Core Lab at
St. Jude Children's Research Hospital) for 8 h, washed twice with
1XPBS, and cultured for 3 days. To quantify the population of cells with
active caspases 3 and 7, cells were incubated at 37 °C with recon-
stituted FAM-FLICA® at a 1:300 dilution (ImmunoChemistry Technol-
ogies 94) for 30 min. Cells were fixed in a 4% formaldehyde solution at
room temperature for 15 min and washed twice with 1X PBS. FAM-
FLICA−positive cells were quantified by FACS (Excitation: 492 nm,
Emission: 520 nm). FACS data were analyzed by FlowJo. To examine
whether cell death is via activation of caspase 8 or 9, Z-IETD-FMK (a
caspase 8 inhibitor) and Z-LEHD-FMK TFA (a caspase 9 inhibitor) were
dissolved in DMSO at 50 mM (Compound Management Center at St.
Jude Children's Research Hospital). After cells were incubated with
mCherry-Cre lentivirus for 8 h, these compounds were added at a
series of concentrations and incubated for 3 days before FACS analysis.
For all the inhibitor treatment assays, the medium with inhibitors was
changed every 2 days. The gating strategy and representative FACS
plots are shown in Supplementary Information (Supplementary
Fig. 4e, Supplementary Fig. 14).

### Subcellular protein extraction

After washing cells once with 1x PBS, they were resuspended in 2x
volume of Buffer A supplemented with PI, 1 mM DTT, and 0.1% TritonX-
100 and placed on ice for 5 min. Cells were centrifuged at $1750 \times g$ for
for 2 min at 4 °C and supernatant was collected as the cytoplasmic
fraction. The nuclear pellet was then resuspended in 1x volume of
Buffer D supplemented with PI, 1 mM DTT, and 0.1% TritonX-100 and

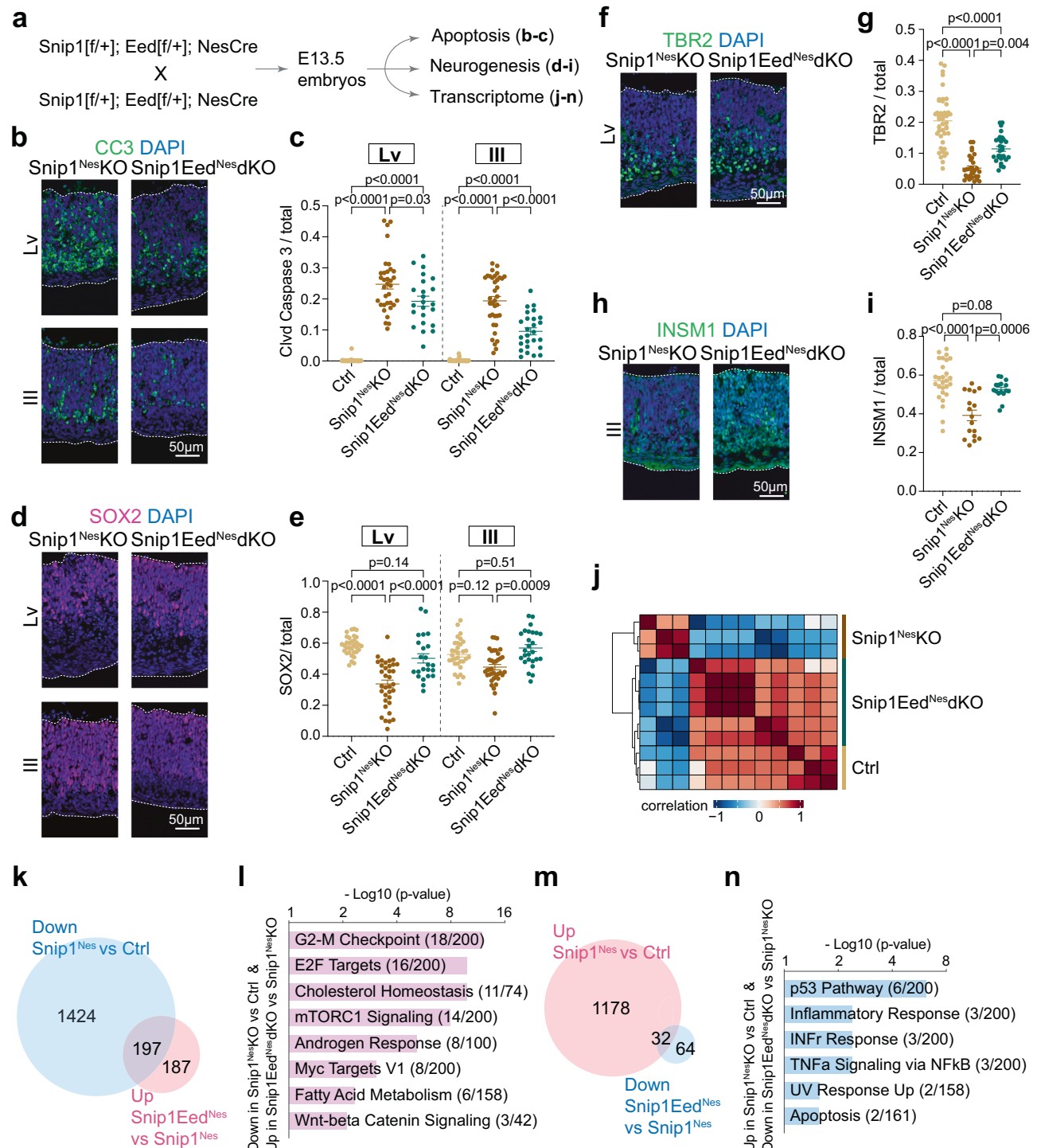

**Fig. 6 | EED depletion reduces apoptosis and brain dysplasia of the Snip1[Nes]-KO brain. a** Schematic of the genetic cross depleting both SNIP1 and EED for downstream assays. **b**, **c** IF analysis of CC3 overlayed with DAPI of the E13.5 brain. Bar, 50 μm. Each data point represents one image. Eight control embryos, 7 *Snip1[Nes]*-KO embryos, and 5 *Snip1[Nes]-Eed[Nes]*-dKO embryos were analyzed. For lateral ventricle, n = 38 images (control), n = 34 (*Snip1[Nes]*-KO), and n = 23 (*Snip1[Nes]-Eed[Nes]*-dKO). For third ventricle, n = 38 (control), n = 36 (*Snip1[Nes]*-KO), and n = 26 (*Snip1[Nes]-Eed[Nes]*-dKO). **d**–**i** IF of NPC marker SOX2, and intermediate progenitor markers TBR2 and INSM1 overlayed with DAPI of the E13.5 brain. Bar, 50 μm. The populations of (**e**) SOX2-positive, (**g**) TBR2-positive, or (**i**) INSM1-positive cells in the neuroepithelial lining of lateral and/or third ventricles were quantified. Each data point represents one image. In (**g**), n = 43 images (control), n = 30 (*Snip1[Nes]*-KO), and n = 29 (*Snip1[Nes]-Eed[Nes]*-dKO). In (**i**), n = 27 (control), n = 17 (*Snip1[Nes]*-KO), and n = 16 (*Snip1[Nes]-Eed[Nes]*-

dKO). For Panels **c**, **e**, **g**, and **i**, data are presented as mean ± SEM, and two-way ANOVA was used for statistical analysis. **j** Unsupervised clustering of RNA-seq data from control (n = 3), *Snip1[Nes]*-KO (n = 3), and *Snip1[Nes]-Eed[Nes]*-dKO (n = 6) brains at E13.5. RNAs from forebrain and midbrain regions were sequenced and merged for downstream analyses. Blue indicates a negative correlation and red indicates a positive correlation. **k**, **m** Venn diagrams displaying the numbers of differentially expressed genes with FDR < 0.05. The lists of downregulated genes in *Snip1[Nes]*-KO vs. control and upregulated genes in *Snip1[Nes]-Eed[Nes]*-dKO vs. *Snip1[Nes]*-KO are compared in (**k**). The lists of upregulated genes in *Snip1[Nes]*-KO vs. control and downregulated genes in *Snip1[Nes]-Eed[Nes]*-dKO vs. *Snip1[Nes]*-KO are compared in (**m**). **l**, **n** Gene ontology of the rescued genes corresponding to Fig. 6k, m. Genes were searched against Molecular Signatures Database (MSigDB) Hallmark 2020 using Enrichr[56]. Source data are provided in a Source Data file (**c**, **e**, **g**, **i**, **l**, **n**).

placed on ice for 30 min (If volumes were large, tubes were rotated at 4 °C). The lysate was centrifuged at $1750 \times g$ for for 2 min at 4 °C and supernatant was collected and diluted with an equal volume of $H_2O$ (nuclear fraction). The chromatin pellet was washed once with cold 1x PBS and resuspended in 1x volume of 0.1 N HCl at 4 °C overnight (O/N). The supernatant was neutralized with the equal volume of 1.5 M Tris-HCl (pH 8.8) (chromatin fraction). For whole-cell lysate extraction, cells were washed once with 1X PBS and resuspended directly in 2x volume of Buffer D supplemented with PI, 1 mM DTT, and 0.1% TritonX-100.

## Co-Immunoprecipitation

Nuclear proteins were extracted as above. Then, 15 µL of protein A and 15 µL of protein G Dynabeads™ (Invitrogen 10002D and 10004D) were washed once with 1x PBST. For pre-bound co-immunoprecipitation, the beads were resuspended in 100 µL of HEPM and 4 µg primary antibody was added. The tube was gently shaken at room temperature for 2 h. The beads were washed once with 1x PBST, and approximately 2.5 mg of nuclear extract was added to the antibody-prebound beads and the tube was rotated at 4 °C for 2.5–4 h. Beads were then washed three times with 1x PBST and proteins were eluted with 0.1 M glycine (pH 2.3) at room temperature. Eluates were neutralized with 1/10 volume of 1.5 M Tris-HCl (pH 8.8). For co-immunoprecipitation with free antibodies, approximately 2.5 mg of nuclear extract was incubated with 4 µg of primary antibody and rotated at 4 °C for 4 h. Beads were then added to the extract and gently shaken for 1 h at room temperature. The same washing and elution steps were performed as for the pre-bound co-immunoprecipitation.

## Western blotting (WB)

For SDS-PAGE, resolving and stacking gels were prepared using the following composition. Resolving gels: 6–12% ProtoGel (National Diagnostics EC8901LTR), 0.375 M Tris-HCl (pH 8.8), 0.1% SDS, 0.1% ammonium persulfate (APS) and 0.1% TEMED (National Diagnostics EC-503). Stacking gels: 3.9% ProtoGel, 0.125 M Tris-HCl (pH 6.8), 0.1% SDS, 0.05% APS, and 0.12% TEMED. After proteins were separated by SDS-PAGE, they were transferred onto a 0.45 µm nitrocellulose membrane (Bio-Rad 1620115) by the semi-dry transfer system. Membranes were blocked with 2% BSA in HEPM for 1 h at room temperature and incubated in primary antibodies diluted in the 2% BSA at 4 °C O/N. On the following day, the membrane was washed three times with 1x PBST and incubated in IRDye®-conjugated secondary antibodies (LI-COR) or Clean-Blot™ IP detection reagent (ThermoFisher 21230) on a shaker for 1 h at room temperature. The membrane was washed three times with 1x PBST and immediately imaged on an Odyssey® Fc imaging system (LI-COR). The membrane stained with Clean-Blot™ IP detection reagent was treated with SuperSignal™ West Pico PLUS Chemiluminescent Substrate (ThermoFisher 34577) for at least 5 min at room temperature before imaging. Signals were quantitated using the Image Studio™ software (version 1.0.14; LI-COR).

## BrdU administration

Mice were administered 5-bromo-2′-deoxyuridine (BrdU, Sigma-Aldrich B5002) reconstituted in sterile 1x PBS by intraperitoneal injection at a dose of 50 mg/kg. After 5 h, the mice were sacrificed for dissection.

## Cryosection

Mouse embryos at an indicated embryonic day were fixed in 4% formaldehyde at 4 °C O/N. The embryos were washed three times in 1x PBS for 30 min at room temperature and placed in 15% sucrose diluted in 1x PBS at room temperature until the embryos sank to the bottom of the tube. Then, the embryos were moved to a 30% sucrose solution and incubated at room temperature until the embryos sank. The embryos were then treated in the embedding medium Tissue-Tek® O.C.T.

Compound (Sakura Finetek USA INC 4583) to rinse the residual sucrose. Each embryo was mounted in the embedding media in a cryosection mold placed on dry ice and 12 µm sagittal cryosections were obtained (Leica CM3050 S).

## Immunofluorescence (IF)

For cryosections, slides were permeabilized in 1x PBST at 4 °C O/N. After drawing the outline of the staining area with a hydrophobic barrier pen (ImmEdge® H-4000), slides were blocked in the IF blocking buffer for 2-3 h at room temperature. Primary antibodies were diluted at an optimized concentration in the IF-blocking buffer and incubated at 4 °C O/N. Sections were washed three times with 1x PBST and incubated with the fluorescent dye–conjugated secondary antibodies (Invitrogen) diluted at 1:500 for 2-3 h at room temperature. Sections were washed three times with 1x PBST and incubated in 1 mg/mL DAPI (Sigma-Aldrich D9542) diluted at 1:500 in 1x PBS at room temperature for 1 h. Finally, sections were washed once with 1x PBS and mounted with ProLong™ Gold Antifade Mountant (Life Technologies P10144). For detecting BrdU, tissues were permeabilized as above and rinsed with 1x PBS for 5 min at room temperature. Then, tissues were treated with 2 N HCl for 1 h at room temperature. Sections were washed multiple times with 1x PBS to remove all traces of HCl and were blocked and stained as above. Images were acquired with a Nikon C2 laser scanning confocal microscope.

## Image analysis

All of the IF image analyses were performed by FIJI. For counting the number of cells, the images were first converted to an 8-bit grayscale. In the automatic nuclei counter plugin (ITCN) of FIJI, for each primary antibody, "Width", "Minimum Distance" and "Threshold" were set manually based on the area and the intensity of the signal on the representative images. The same parameters on ITCN were applied for control and experimental groups. For each group, at least five images were analyzed. For measuring the thickness of a tissue, distance between two hand-drawn lines on an image was measured. The publicly available InteredgeDistance macro was used for calculating the average of the shortest distances of randomly selected points on the two lines. For counting the GABA-positive TUJ1-positive cells, the GABA images were processed through thresholding before overlayed with the TUJ1 images. By using the TUJ1 signals as the outline of each neuron, the number of neurons with at least one GABA punctum was manually counted.

## iDISCO

We performed iDISCO clearing method using the protocol developed by the Tessier-Lavigne lab[95]. In brief, mouse embryos at an indicated embryonic day were fixed in 4% formaldehyde and washed in 1X PBS (see Cryosection section). Samples were first dehydrated by incubating them for 1 h each with the increasing concentrations of methanol at 20%, 40%, 60%, 80%, and 100% at room temperature. Samples were then incubated in 66% dichloromethane (DCM)/ 33% methanol O/N. Samples were washed twice in 100% methanol, pre-chilled at 4 °C, and incubated with 5% $H_2O_2$ in methanol at 4 °C O/N. Samples were then rehydrated by incubating them for 1 h each with the decreasing concentrations of methanol at 80%, 60%, 40%, 20% and 1X PBS at room temperature. Samples were washed twice in PTx.2 for 1 h each. For immunolabeling, samples were incubated in a permeabilization solution at 37 °C for 2 days and blocked in Blocking solution at 37 °C for 2 days. Primary antibodies diluted in PTwH supplemented with 5% DMSO and 3% donkey serum were incubated at 37 °C for 3-4 days. Samples were moved to the freshly diluted antibodies and incubated for another 3–4 days. Samples were washed in PTwH for 4-5 times for the entire day and incubated with the secondary antibodies diluted in PTwH supplemented with 3% donkey serum at 37 °C for 3-4 days. Again, the antibodies were replaced with freshly diluted antibodies and

incubated for an additional 3-4 days. Samples were washed in PTwH for 4-5 times for the entire day. For clearing, samples were first dehydrated with methanol as described above and treated with 66% DCM/ 33% methanol for 3 h at room temperature. Samples were washed twice with 100% DCM to rinse off methanol and incubated with dibenzyl ether to clear the tissues. Images were acquired with a LaVision light sheet microscope.

## RNAscope VS duplex assay

To detect the expression pattern of transcripts in the brain, mouse embryos at an indicated embryonic day were fixed in 10% neutral-buffered formalin (NBF) at room temperature. Fixed embryos were paraffin-embedded and sectioned at a thickness of 4 μm. RNA probes were designed and purchased from ACDBio; *Snip1* probe targeting 1270-2274 bp of NM_001356560.1 and *Eomes/Tbr2* probe targeting 1289-2370 bp of NM_010136.3 (Cat. 429649-C2). Sectioning and in situ hybridization (ISH) were done by the Comparative Histology Core at St. Jude Children's Research Hospital by following manufacturer's instructions. Brightfield images were acquired with Keyence BZ-X700.

## RNA extraction and reverse transcription

Total RNA was extracted from FACS-sorted cells using TRIzol reagent (Invitrogen™ 15596026) and Direct-zol™ RNA Microprep (Zymo Research R2062) by following manufacturer's instructions. DNA digestion with DNase I was also performed as part of the RNA extraction. cDNA was prepared with 500–1000 ng of total RNA using SuperScript™ IV VILO™ Master Mix (ThermoFisher 11766050) by following manufacturer's instructions.

## Real-time quantitative PCR (RT-qPCR)

RT-qPCR was performed with PowerUp™ SYBR™ Green Master Mix (Applied Biosystems™ A25778) using Applied Biosystems QuantStudio 3. Primers are listed in Supplementary Table 3. Three technical replicates were set up for each gene target. For data analysis, $2^{-\Delta\Delta Ct}$ method, which compares the difference in the threshold cycle values of control and experimental samples, was used. The threshold cycle values of a gene of interest was normalized to that of housekeeping gene *Gapdh* or *βActin*.

## RNA-Seq analysis

Paired-end 100 bp sequencing was performed on NovaSeq6000 sequencer by following the manufacturer's instructions (Illumina). Raw reads were first trimmed using TrimGalore (version 0.6.3) available at: https://www.bioinformatics.babraham.ac.uk/projects/trim_galore/, with parameters '--paired --retain_unpaired'. Filtered reads were then mapped to the *Mus musculus* reference genome (GRCm38.p6 + Gencode-M22 Annotation) using STAR (version 2.7.9a)[96]. Gene-level read quantification was done using RSEM (version 1.3.1)[97]. To identify the differentially expressed genes between control and experimental samples, the variation in the library size between samples was first normalized by trimmed mean of $M$ values (TMM) and genes with CPM < 1 in all samples were eliminated. Then, the normalized data were applied to linear modeling with the voom from the limma R package[98]. Gene set enrichment analysis (GSEA) was performed against the MSigDB database (version 7.1), and differentially expressed genes were ranked based on the their $\log_2(FC) * -\log_{10}(p\text{-value})$[99,100].

## CUT&RUN

Approximately $3 \times 10^5$ NPCs sorted for *Sox2*-eGFP were mixed with $3 \times 10^4$ *Drosophila* S2 cells per reaction. We performed CUT&RUN using the protocol developed by the Henikoff lab[51]. In brief, Bio-Mag®Plus Concanavalin-A (Con A) coated beads (Bangs Laboratories BP531) were washed and activated with Binding buffer. Nuclei of NPCs with S2 spike-in were gently prepared (see Subcellular Protein

Extraction section). Activated Con A beads and nuclei were then mixed and rotated for 5 min at room temperature. They were then blocked with Digitonin block buffer for 5 min at room temperature. 0.5–1 μg of primary antibody with 0.25 μg Spike-in antibody (Active Motif 61686) diluted in Digitonin block buffer was added to the bead-nuclei mixture and incubated for 3 h (histone marks) or O/N (the rest) at 4 °C. Beads were washed three times with Digitonin block buffer and incubated with pA-MNase for 1 h at 4 °C. Beads were washed three times with Wash buffer and incubated in Wash buffer for 10 min on ice. The pA-MNase was activated by incubating the beads with 2 mM $CaCl_2$ for 25 min on ice and quenched by adding the Stop buffer. DNA was released from the beads by incubating them for 30 min at 37 °C and collected by centrifugation at 16,000 x $g$ for 5 min at 4 °C. DNA was then isolated by using a phenol/chloroform extraction method. Libraries were constructed using ACCEL-NGS® 1 S Plus DNA Library Kit by following the manufacturer's instructions (Swift Biosciences 10024). Purified libraries were analyzed with TapeStation (Agilent), using the High Sensitivity D1000 reagents (Agilent 5067-5585) before sequencing. IgG primary antibody was used as the negative control.

## CUT&RUN-reChIP (Chromatin immunoprecipitation)

We first performed CUT&RUN with anti-SNIP1 antibody using approximately $7 \times 10^6$ cultured NPCs cells per reaction. Before proceeding to ChIP, Dynabeads™ were washed once with 1x PBST and incubated with 2 μg of primary antibody on the vortex (setting 3) for 2 h at room temperature. The eluates from CUT&RUN were then added to the antibody-bound Dynabeads™ and incubated for 3 h at 4 °C while rotating. Dynabeads™ were washed twice with 1x PBST. Chromatin was then eluted by incubating in ChIP elution buffer for 15 min at 65 °C. The eluate volume was adjusted to 100 μL before proceeding to DNA isolation using MinElute PCR Purification Kit (Qiagen 28004). Libraries were constructed as were done with CUT&RUN samples using ACCEL-NGS® 1S Plus DNA Library Kit. IgG primary antibody was used as the negative control for ChIP.

## CUT&RUN analysis

CUT&RUN and CUT&RUN-reChIP libraries were sequenced on NovaSeq6000 sequencer and generated 50 bp paired-end reads. The reads were aligned to *Mus musculus* mm10 genome reference and *Drosophila melanogaster* dm6 genome reference by BWA (version 0.7.170.7.12, default parameter). Duplicated reads were marked by the bamsormadup from the biobambam tool (version 2.0.87) available at https://www.sanger.ac.uk/tool/biobambam/. Uniquely mapped reads were kept by samtools (parameter "-q 1 -F 1804," version 1.14). Fragments <2000 bp were kept for peak calling and bigwig files were generated for visualization. SICER[52] and MACS2[53] were both used for peak calling, to identify both the narrow and broad peak correctly. With SICER, we assigned peaks that were at the top 1 percentile as the high-confidence peaks and the top 5 percentile as the low-confidence peaks. Two sets of peaks were generated: strong peaks called with parameter 'FDR < 0.05' by at least one method (MACS2 or SICER) and weak peaks called with parameter 'FDR < 0.5' by at least one method (MACS2 or SICER). Peaks were considered reproducible if they were supported by a strong peak from all replicates or at least one strong peak and a weak peak in the other replicates. For differential peak analysis, peaks from two replicates were merged and counted for the number of overlapping extended reads for each sample (bedtools v2.24.0)[101]. Then we detected the differential peaks by the empirical Bayes method (eBayes function from the limma R package)[98]. Peaks were annotated based on Gencode following this priority: "Promoter. Up": if they fall within TSS−2 kb, "Promoter.Down": if they fall within TSS −2 kb, "Exonic" or "intronic": if they fall within an exon or intron of any isoform, "TES peaks": if they fall within TES ± 2 kb, "distal5" or "distal3" if they are with 50 kb upstream of TSS or 50 kb downstream

of TES, respectively, and "intergenic" if they do not fit in any of the previous categories. For downstream analyses, heatmaps were generated by deepTools[102] and gene ontology was performed with Enrichr[56,103] and GSEA, in addition to custom R scripts.

## Integrative analysis of RNA-seq and CUT&RUN

To identify differential peaks correlated with gene expression changes, we adapted some ideas from the intePareto method[104]. For each gene $g$, we converted its RNA-seq $\log_2(FC)$ to a z-score by scaling the $\log_2(FC)$ to the standard deviation of all fold-changes in the sample using the following formula:

$$z_g = \frac{log_2FC(g)}{sd(log_2FC)} \quad (1)$$

Instead of associating a single peak to each gene as was done in the original intePareto method, we associated a gene to all peaks located within TSS ± 50 kb to be able to unbiasedly identify the most correlated peak. Similarly, we converted the fold change value of each peak $p$ in $TSS_g \pm 50$ kb to a z-score using the same formula but using ChIP-seq fold change values:

$$z_p = \frac{log_2FC(p)}{sd(log_2FC)} \quad (2)$$

For each gene-peak pair, we calculated a combined z-score by multiplying their z-scores as follows:

$$z_{g,p} = z_g \times z_p \quad (3)$$

The multi-objective Pareto optimization was then calculated using the 'psel' function from the 'rPref' R/package (v1.3) [https://doi.org/10.32614/RJ-2016-054]. The peaks from the top 10 best Pareto levels were selected as the most correlated/anti-correlated.

## Statistics and reproducibility

Statistical analyses were performed using R 4.0.1 or Prism 9.0.2 (GraphPad Software). Parameters of statistical analyses such as the number of replicates and/or experiments ($n$), deviations, $p$-values, and types of statistics tests are included in the figures or figure legends. For all the in vivo experiments, at least three biological replicates were assessed. All in vitro assays were performed with at least two independent sample sets. Error bars on graphs represent the mean ± SEM.

## Reporting summary

Further information on research design is available in the Nature Portfolio Reporting Summary linked to this article.

# Data availability

Source data for this paper are provided. For sequencing data analyses, mm10 (*Mus musculus*) and dm6 (*Drosophila melanogaster*) reference genomes were used. All the data used in this study was deposited in GEO under GSE212445. Source data are provided with this paper.

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

## Acknowledgements

The authors thank V. Shanker for editing the manuscript; J. Houston and K. Lowe for FACS; M. Evans, I. Lam, I. Chapman, H. Chen, and E. Rivera-Peraza for experimental assistance. RNAscope was performed by the Comparative Histology Core at SJCRH. Sequencing was performed at the Harwell Center for Biotechnology and images were acquired at the Cell & Tissue Imaging Center, both of which are supported by SJCRH and NCI P30 (CA021765). M.N.D., Y.F., and B.X. are supported by NCI P30 grant (CA21765). This work is funded by the American Lebanese Syrian Associated Charities, American Cancer Society (132096-RSG-18-032-01-DDC), and NIH (1R01GM134358-01). The content is solely the responsibility of the authors and does not necessarily represent the official views of the National Institutes of Health. The funders had no role in study design, data collection and analysis, decision to publish, or preparation of the manuscript.

## Author contributions

Y.M. most experiments, data analyses, and manuscript writing. M.N.D. and B.X. bioinformatics analyses. N.C. mouse breeding, immunoprecipitation, and Western blotting. P.S. ideas on studying apoptosis and inhibitor assays. K.L.: mouse harvesting and rescue by lentiviral SNIP1. G.W.: SNIP1 variants and supervision of Y.F., B.X., and M.N.D. X.Y.: DNA

construct, qPCR of CUT&RUN, and RT-qPCR. Y.F.: supervision of bioinformatics analyses. J.C.P.: project design, data analyses, and manuscript writing with inputs from all authors.

## Competing interests

The authors declare no competing interests.
