## [Peer Review File · Nature Communications]

SNIP1 and PRC2 coordinate cell fates of neural progenitors during brain developmentREVIEWER COMMENTS

Reviewer #1 (Remarks to the Author):

Yurika Matsui and colleagues present data reveals a novel epigenetic pathway critical for coordinating survival, self-renewal, and differentiation of stem cells in the developing brain. Here the authors Used embryonic mouse brains as a model, uncovered that Snip1 promotes neurogenesis and cortical development and suppresses intrinsic apoptosis and further showed Snip1 and PRC2 co-occupy chromatin targets to co-regulate the selected genetic programs. And the rescue of Snip1Nes-KO NPCs by Eed/PRC2 depletion suggested that Snip1 and PRC2 have a balancing relationship for regulating genetic programs in the brain. These findings about the Snip1-PRC2 interactions in NPC survival and functions improve the understanding about the neurodevelopmental disorders caused by SNIP1 mutations and PRC2 dysfunction. These results suggest a loci-dependent regulation of PRC2 and H3K27 marks to toggle between stem cell fates. Overall, this is a interesting story.

The research showed plenty of phenotypes and data analysis related in SNIP1-PRC2 in NPCs, however, the quality of the data in the article need improvements. This connection between SNIP1 and PRC2 and regulation of cortical development, however, it is not very strong and needs to be strengthened by additional experiments prior to publication. If the following problems are well-addressed, this reviewer believes that the essential contribution of this paper are important for neurodevelopmental disorders.

1. The title of the article is not well fit. The title should be summary and abstractive. "stem cell" in the title seems to be an inaccurate statement. The research is mainly focused on "neural progenitor cells".
2. The author performed RNA-seq of SOX2-positive NPCs and SOX2-negative differentiated cells. Figure 2a showed FACS for GFP-positive cells, indicating the GFP-positive cells are SOX2-positive NPCs, I was wondering how to define SOX2-negative differentiated cells (Line 155). The author should explain the results. Another question, why did the authors compare SOX2-negative differentiated cells with SOX2-positive cells?
3. Figure 4, the most interesting result is the interaction between Snip1 and PRC2. Here I have several questions 1) How about the PRC2 and the histone levels after Snip1 deletion? Did they have changes? 2) PRC2 mainly trimethylates lysine 27 of histone H3 and is composed of three essential core subunits: EZH2, EED and SUZ12. Why do you use Nes: Cre to excise exons 3 to 6 of Eed (a PRC2 core subunit) to generate in EedNes-KO and Snip1Nes-EedNes-Dko, but not EZH2 or SUZ12? 3) Why had no Co-IP result between Snip1 and EED in Fig.4?
4. The authors performed a lot of RNA-seq, Fig2 and Fig5, however, they didn't show further confirmation of these potential dysregulated targets with RT-PCR or WB.
5. Fig 1a and Supplementary Fig. 1c: WB of control and Snip1Nes-KO brains at E13.5 to verify the expression of SNIP1 using anti-Snip1 antibody from ProteinTech or anti-Snip1 antibody from ThermoFisher. However, WB results still showed bands in the Snip1Nes-KO group using the above two antibodies. And the statistics of the immunoblot results need to be provided. Figure4 a,b,c which one is the band of snip1, it seems different from Figure1a.
6. On line 152 and Supplementary Fig. 5e-f: Why overexpress human SNIP1, although it shares 85% identity with mouse SNIP1. overexpression of mouse SNIP1 may be a better option.
7. The introduction to this paper described that "Global knockout (KO) of Snip1 in zebrafish embryos causes reduction in GABAergic and glutamatergic neurons". However, the results in this paper suggested the relative thickness of the Tuj1-positive region did not significantly differ between the Snip1Nes-KO and control at E13.5 said on line 100. How does the author explain the discrepancy between this phenotype and that of zebrafish. Does global knockout (KO) of Snip1 lead to variation of GABAergic and glutaminergic neurons in mouse embryos?
8. One more confusing question, the results showed that there was a high percentage of cell apoptosis and a decrease of SOX2-positive cells, why cortex thickness of Tuj1+ didn't change?
9. Actually, EED cKO in forebrain also leads to high apoptosis (PMID: 35931079), why snip1Nes-EedNes-Dko could rescue apoptosis?
10. The full name of Snip1 should be given when it first appears in the paper. And the same is true for other genes.
11. The expression of the gene SNIP1 is not uniform in case. For example, line 57 is "Snip1", and line 57 is "SNIP1". Please check the names of other genes.
12. Line 254-258, the 32 upregulated genes whether include the 9 genes (Supplementary Figure 7j)?

There are many representative figures are not enough for the journal:

13. Supplementary Fig. 10C: IF analysis of Sox2 and CC3 in E13.5 brain in this figure. However, in this diagram, I don't find staining result of CC3.
14. Supplementary Figure 2 e and g, why the representative picture is not consistent in the same position?
15. Figure 1 h-i, Supplementary Figure 3 f-h, Supplementary Figure 4 e-g, Supplementary Figure 5a, the statistics of the immunostaining results need to be provided.
16. In figure 5f, the DAPI exposure values of the two groups were inconsistent.
17. In Supplementary figure 4h, the location of the brain appeared to be inconsistent between the control and Snip1NesKO group, and the expression of Otx2 and foxg1 was demonstrated to be different between the two groups.
18. Supplementary Fig. 7 Profiling of H3K27me3 and H3K27ac in control and Snip1Nes-KO NPCs, the second H3K27me3 should be H3K27ac.

Reviewer #2 (Remarks to the Author):

Here, Matsui et al reveal a role for Snip1 in suppressing intrinsic apoptosis genes and promoting proliferation in NPCs. They show that it cooperates with PRC2 complex and regulates H3K27 mark turnovers. The data has been tested using both in vivo (KO mice) and in vitro models with a range of biochemical and cellular assays. This study provides interesting insights into the regulation of apoptosis during cortical development, a key process during embryogenesis. However, authors need to address several concerns before this study is considered in Nature Communications.

Comments:

1. It would be good to perform BrDU labelling to compare proliferative content in KO versus WT mice.
2. The claim that Snip1 recruits PRC2 needs to be substantiated by:
 - Depletion of Snip1 and analysis of the kinetics of PRC2 components
 - ChIP-ReChIP to confirm their co-occupancy on chromatin.
3. Line No 74. Maybe it is Snip1, not Snip11.
4. In Figure 2h increasing the concentration of Z LEHD-FMK does not show any effect on intrinsic apoptosis inhibition. Could authors explain this?
5. What is the intrinsic apoptosis status in in vitro cultured Snip1Nes-KO NPC cells? Can the induction of apoptosis in Snip1Nes-KO NPC cells be reverted in by overexpressing Snip1?
6. Line No 173. Maybe it is Snip1Nes-KO not Snip1Ne-KO.
7. In the section "PRC2 requires Snip1 for localization to chromatin and H3K27me3 deposition" Co-immunoprecipitation of Snip1 with PRC2 subunits (Jarid2, Suz12, and Ezh2) in Snip1Nes-KO mutants would have been an interesting negative control.
8. One of the key questions is how Snip1 is specifically targeted to the different classes of target loci?
9. Could authors rule out whether the actual role of SNIP1 is keeping a balance between proliferating and differentiating progenitors?
10. Is there any overlap between the genes identified in Suz12 and Ezh2 CUT & RUN assay and the genes that were downregulated in Snip1Nes-KO mutants? If there is minimal or no overlap between these two datasets, it will suggest that Snip1 binds independently of PRC2 at neurodevelopmental gene loci and in combination with PRC2 at intrinsic apoptosis pathway gene loci.
11. Have authors explored whether any SNPs exist in SNIP1 in disease databases in particular for brain disorders?
12. There is a lack of data/observations to support the statement at lines 28-29
13. Authors may investigate and include the detailed observations on intrinsic properties mentioned in Bernard C, Exposito-Alonso D, Selten M, et al. Cortical wiring by synapse type-specific control of local protein synthesis. Science. 2022;378(6622):eabm7466. doi:10.1126/science.abm7466

We appreciate the reviewers for their thoughtful comments, which collectively have much improved all sections of this manuscript. We have obtained new data to strengthen the following conclusions.

- (a) In addition to intrinsic apoptosis, SNIP1 suppresses neural differentiation programs in differentiating cells. Once cells commit to differentiation, SNIP1 promotes the specification of the GABAergic neuronal lineage.
- (b) As suggested by Reviewer 2: independent of PRC2, SNIP1 binds target genes to promote key genetic programs for neurodevelopment.
- (c) Discovery of TGF β and NF κ B signaling pathways for regulating SNIP1 localization to target genes. Rigorous CUT&RUN data analysis combined with treatment of 3 inhibitors to TGF β and NF κ B signaling enable us identify the target genes regulated by SNIP1 and the key signaling pathways.
- (d) Improved characterization of the PRC2–SNIP1 interaction on chromatin by using the innovative CUT&RUN-reChIP assay.
- (e) As suggested by Reviewer 1: discovery of *Cdkn1a* as a key downstream effector of PRC2 and SNIP1 in regulating intrinsic apoptosis in neural progenitor cells (NPCs).

In the revised text, the citations were changed to Nature journal style, and text changes were tracked.

Reviewer #1 (Remarks to the Author):

Yurika Matsui and colleagues present data reveals a novel epigenetic pathway critical for coordinating survival, self-renewal, and differentiation of stem cells in the developing brain. Here the authors Used embryonic mouse brains as a model, uncovered that *Snip1* promotes neurogenesis and cortical development and suppresses intrinsic apoptosis and further showed *Snip1* and PRC2 co-occupy chromatin targets to co-regulate the selected genetic programs. And the rescue of *Snip1*^{Nes}-KO NPCs by *Eed*/PRC2 depletion suggested that *Snip1* and PRC2 have a balancing relationship for regulating genetic programs in the brain. These findings about the *Snip1*–PRC2 interactions in NPC survival and functions improve the understanding about the neurodevelopmental disorders caused by SNIP1 mutations and PRC2 dysfunction. These results suggest a loci-dependent regulation of PRC2 and H3K27 marks to toggle between stem cell fates. Overall, this is a interesting story. The research showed plenty of phenotypes and data analysis related in SNIP1-PRC2 in NPCs, however, the quality of the data in the article need improvements. This connection between SNIP1 and PRC2 and regulation of cortical development, however, it is not very strong and needs to be strengthened by additional experiments prior to publication. If the following problems are well-addressed, this reviewer believes that the essential contribution of this paper are important for neurodevelopmental disorders.

We agree that the connection between SNIP1 and PRC2 to cortical development indeed lacks strong data support. We had attempted experiments to examine a direct role of SNIP1 over cortical development. However, massive death of *Snip1*^{Nes}-KO NPCs renders *in vivo* studies of cortical

development beyond E13.5 impossible. Thus, we reduced discussion about SNIP1's influence over cortical development in lines 23-24, 25-26, 74, 118, 190, and 267 in the initial submission.

Separately, to address Reviewer 2's point 11, in lines 62-64 of the revised text, we add that we identified a pathogenic variant of SNIP1, "R111C, that is significantly associated with epilepsy and skull dysplasia (Supp Fig 1a). These data suggest a critical role of SNIP1 in human neurodevelopment."

Supp Fig 1a. Graphical summary of *SNIP1* variants that are associated with epilepsy and skull dysplasia. PMID's of the source data are listed for the unpublished variants. The R111C variant passed statistical threshold, whereas the other 3 variants nearly passed statistical threshold.

1. The title of the article is not well fit. The title should be summary and abstractive. "stem cell" in the title seems to be an inaccurate statement. The research is mainly focused on "neural progenitor cells".

The title of this manuscript has changed to "Snip1 and PRC2 coordinate cell fates of neural progenitors during brain development" (line 1).

2. The author performed RNA-seq of SOX2-positive NPCs and SOX2-negative differentiated cells. Figure 2a showed FACS for GFP-positive cells, indicating the GFP-positive cells are SOX2-positive NPCs, I was wondering how to define SOX2-negative differentiated cells (Line 155). The author should explain the results. Another question, why did the authors compare SOX2-negative differentiated cells with SOX2-positive cells?

We appreciate this opportunity to clarify this point, which also relates to Reviewer 1's point 8. We changed lines 157-160 to "Therefore, we aimed to examine the molecular effect of SNIP1 in differentiating neural cells, which have lost the expression of NPC marker SOX2. We performed RNA-seq of SOX2:GFP-negative cells sorted from E13.5 *Snip1^{Nes}-KO* and sibling control brains (Fig 2a, Supp Fig 4a)."

Respectfully, we want to mention that we did not compare SOX2-negative differentiated cells with SOX2-positive cells.

We then wondered perhaps the reviewer meant to ask "why did the authors [not] compare SOX2-negative differentiated cells with SOX2-positive cells?" Below, we briefly outline the results from the (1) and (2) comparisons.

- (1) WT SOX2:GFP-positive versus negative cells.
- (2) *Snip1^{Nes}-KO* SOX2:GFP-positive versus negative cells.

We considered whether *Snip1^{Nes}-KO* alter genetic programs by using fold-change >2 and p <0.05 to identify differentially expressed genes in (1) and (2) pairwise comparisons (Fig 2-pt 2a, b). This identified genes that had higher (587 red underlined, Fig 2-pt 2c) or lower (1512 blue underlined, Fig 2-pt 2d) expression only in *Snip1^{Nes}-KO* SOX2:GFP-positive cells. Gene ontology analysis revealed

the enriched pathways in higher (Fig 2-pt 2e) or lower (Fig 2-pt 2f) only in *Snip1^{Nes}*-KO SOX2:GFP-positive cells. We infer from the data that SNIP1 is required to maintain the expression of these genes in NPCs compared to differentiated cells.

Figure legends for Re1-2. Principal component analysis of (a) WT and (b) *Snip1^{Nes}*-KO SOX2:GFP-positive versus negative cells. The number of differentially expressed genes were identified by the criterion of FDR<0.05. Genes that were expressed concordantly (c) higher or (d) lower in WT and *Snip1^{Nes}*-KO SOX2:GFP-positive versus negative cells. Enriched gene ontology terms in genes that had significantly (e) higher and (f) lower expression only in *Snip1^{Nes}*-KO SOX2:GFP-positive. *P* values were listed along the terms.

3. Figure 4, the most interesting result is the interaction between Snip1 and PRC2. Here I have several questions

We thank the Reviewer for this assessment.

3-1) How about the PRC2 and the histone levels after Snip1 deletion? Did they have changes?

We first used RNA-seq to compare the transcript levels (transcript per million, TPM, values, y-axis) of PRC2 core subunits in WT and *Snip1^{Nes}*-KO SOX2:GFP-positive NPCs. SNIP1 suppresses the EZH2 transcripts.

Supp Fig 10d. Transcript levels of PRC2 core subunits in SOX2-positive NPCs. TPM (transcript per million) of each gene is shown. Data are presented as mean ± SEM, and two-way ANOVA was used for statistical analysis. ns = not statistically significant; ****p <0.0001.

We used Western blotting to compare the protein levels of PRC2 subunits and H3K27me3 and H3K27ac in WT and *Snip1^{Nes}*-KO E13.5 brains. This revealed no change in total protein levels between *Snip1^{Nes}*-KO E13.5 brains. In lines 255-257, we added “We tested whether SNIP1 alters the expression of PRC2 subunits. Although SNIP1 depletion lowered EZH2 transcript levels, it did not alter the protein levels of PRC2 subunits, H3K27me3, or H3K27ac (Supp Fig 10d-g).”

Supp Fig 10. e-f WB of control and *Snip1^{Nes}*-KO brains at E13.5. **g** Quantification of WB blots shown in Supp Fig 10e-f. The band intensities of SUZ12, EZH2, and SNIP1 were normalized to that of ACTB and the band intensities of H3K27me3 and H3K27ac were normalized to that of Histone H3. Data are presented as mean ± SEM, and two-way ANOVA was used for statistical analysis. ns = not statistically significant; ***p <0.001.

3-2) PRC2 mainly trimethylates lysine 27 of histone H3 and is composed of three essential core subunits: EZH2, EED and SUZ12. Why do you use Nes: Cre to excise exons 3 to 6 of Eed (a PRC2 core subunit) to generate in EedNes-KO and Snip1Nes-EedNes-Dko, but not EZH2 or SUZ12?

We appreciate the opportunity to clarify our reasoning. A mouse strain containing *Suz12^[fl/fl]* is not available as far as we searched in the mouse stock centers and in the literature. The enzymatic subunits of PRC2, EZH2 and EED, are both well expressed in NPCs, as shown by our RNA-seq profiling (Supp Fig 10d, above). Another study (Henriquez, et al, Mol Cell Neurosci 2013, <https://www.sciencedirect.com/science/article/pii/S104474311300078X?via%3Dihub>) also supports the expression of EZH2 and EED in NPCs.

As EZH2 and EZH1 are well expressed and have redundant function in NPCs, depletion of PRC2 complex would require both EZH2 and EZH1 floxed alleles. Furthermore, *Snip1^{Nes}-KO* lowered EZH2 and did not alter EZH1, EED, or SUZ12 transcripts (Supp Fig 10d, above). These reasons led us to use the *Eed*-floxed allele.

3-3) Why had no Co-IP result between Snip1 and EED in Fig.4?

We have now added EED IP in the revised Fig 5d. We had tried repeatedly to obtain EED bands with SNIP1 IP, but not successfully. We suspect that the antibody to SNIP1 may interfere with the efficient co-IP of EED.

Fig 5d Co-immunoprecipitation followed by WB to examine the interaction between SNIP1 and PRC2. EED was immunoprecipitated in the NPC nuclear extract. RBBP5 was a negative control.

4. The authors performed a lot of RNA-seq, Fig2 and Fig5, however, they didn't show further confirmation of these potential dysregulated targets with RT-PCR or WB.

We appreciate this comment to help us improve. We performed RT-qPCR to validate RNA-seq results. We also used qPCR to validate some of new results throughout the manuscript, such as SNIP1 CUT&RUN-reChIP.

Supp Fig 4c-d. Quantitative PCR of the representative genes that were (c) upregulated or (d) downregulated in the *Snip1^{Nes}-KO* NPCs versus control NPCs. The Cq values of each gene were normalized to those of a housekeeping gene *Actb*. Then, gene expression level in *Snip1^{Nes}-KO* NPCs was normalized to that in control. Data are presented as mean \pm SEM, and multiple unpaired t-tests were used for statistical analysis. *p <0.05; **p <0.01; ***p <0.001; ****p <0.0001.

5. Fig 1a and Supplementary Fig. 1c: WB of control and Snip1Nes-KO brains at E13.5 to verify the expression of SNIP1 using anti-Snip1 antibody from ProteinTech or anti-Snip1 antibody from ThermoFisher. However, WB results still showed bands in the Snip1Nes-KO group using the above two antibodies. And the statistics of the immunoblot results need to be provided. Figure4 a,b,c which one is the band of snip1, it seems different from Figure1a.

We appreciate this opportunity to clarify the SNIP1 WB results. Below, we have provided the original uncropped WB images for SNIP1. Throughout the study, we had used different lots of the anti-SNIP1 antibodies from Thermo Fisher and Protein Tech. Protein Tech antibody was used for Fig 1a and Supp Fig 1d, whereas Thermo Fisher antibody was used for Fig 4a-c. Our rationale was that Protein Tech antibody worked for IP, so Thermo Fisher was used for WB detection. The specificity of

the Protein Tech antibody was shown by Fig 1a and Supp Fig 1d, whereas the specificity of the Thermo Fisher antibody was shown by CUT&RUN using control and *Snip1^{Nes}-KO* NPCs. Statistics of the Fig 1a and Supp Fig 1d results are in Supp Fig 1e. For IP-WB, we regret that could not provide statistics. On Fig 4c, asterisk (*) indicates the SNIP1 protein band.

6. On line 152 and Supplementary Fig. 5e-f: Why overexpress human SNIP1, although it shares 85% identity with mouse SNIP1. overexpression of mouse SNIP1 may be a better option.

We agree with the Reviewer. Mouse and human SNIP1 are conserved by 83.3% (330/396) identity and 89.4% (354/396) homology. Our original intention was to use human SNIP1 cDNA to show that the role of SNIP1 is conserved, and we have now clarified in line 145 “suggesting functional conservation of SNIP1 between humans and mice.”

Our intention is to use mouse SNIP1 for rescue experiments as we did for the original Supp Fig 5e-f. We were allowed for three months for this major revision. We generated the lentiviral construct and then the concentrated lentivirus to deliver mouse SNIP1. But during the 6 weeks until deadline, timed mating of heterozygotes mice did not successfully yield *Snip1^{Nes}-KO* embryos and primary NPCs for the rescue experiments. We apologize that within this time period, we could not obtain this better option. After submission of this revised manuscript, we will be continually performing the planned rescue experiments.

7. The introduction to this paper described that “Global knockout (KO) of *Snip1* in zebrafish embryos causes reduction in GABAergic and glutamatergic neurons”. However, the results in this paper suggested the relative thickness of the Tuj1-positive region did not significantly differ between the *Snip1^{Nes}-KO* and control at E13.5 said on line 100. How does the author explain the discrepancy between this phenotype and that of zebrafish. Does global knockout (KO) of *Snip1* lead to variation of GABAergic and glutaminergic neurons in mouse embryos?

We agree that this is a good point. We analyzed RNA-seq data of differentiating neural cells, which are *Sox2:GFP*-negative populations by FACS. We also performed immunofluorescence of GABA (GABAergic neuronal marker) in control and *Snip1^{Nes}-KO* E13.5 brain. Immunofluorescence for NMDAR2B/GRIN2B (glutaminergic neuronal marker) did not work in E13.5 brains. Except *Gad1* and *Slc6a1*, there was little difference in GABAergic and glutaminergic neuronal transcripts between control and *Snip1^{Nes}-KO*. Immunofluorescence and quantification suggest that GABAergic neuronal specification was lower in *Snip1^{Nes}-KO* E13.5. We note that because of drastic tissue loss in *Snip1^{Nes}-*

KO E13.5 brain, we could not obtain samples beyond E13.5 for robust analysis of subneuronal differentiation. Therefore, we added to lines 178-185, “Because global knockout (KO) of *Snip1* in zebrafish embryos causes reduction in GABAergic and glutamatergic neurons³⁵, we asked whether SNIP1 depletion alters subneuronal lineage specification. Because of drastic brain tissue loss in *Snip1^{Nes}-KO*, we could not robustly analyze brain development beyond E13.5. At E13.5, transcript levels of GABAergic neuronal markers *Gad1* and *Slc6a1* were lower in *Snip1^{Nes}-KO*, and glutamatergic neuronal markers did not differ between *Snip1^{Nes}-KO* and control (Supp Fig 7i-j). Quantification of immunofluorescence showed significantly lower GABA- (GABAergic neuronal marker) positive cells in *Snip1^{Nes}-KO* (Supp Fig 7k-l), which is consistent with data in zebrafish.”

Supp Fig 7. i-j Transcript levels of (f) GABAergic neuronal markers and (g) glutamatergic neuronal markers in SOX2:GFP-negative brain cells. TPM (transcript per million) of each gene is shown. Data are presented as mean \pm SEM, and unpaired t tests were used for statistical analysis. ns = not statistically significant; *p < 0.05; **p < 0.01. **k** IF of a GABAergic neuronal marker GABA overlaid with TUJ1 of the E13.5 brain. Bar, 10 μ m. **l** Quantification of GABA-positive cells that are also TUJ1-positive in the control and *Snip1^{Nes}-KO* brains at E13.5. Cells that are positive for both GABA and TUJ1 in the area of 212.55 μ m² were counted. Each data point represents one 212.55 μ m² image. Data are presented as mean \pm SEM, and two-way ANOVA was used for statistical analysis. ns = not statistically significant; *p < 0.05.

8. One more confusing question, the results showed that there was a high percentage of cell apoptosis and a decrease of SOX2-positive cells, why cortex thickness of Tuj1+ didn't change?

It also surprised us that cortical thickness did not change between control and *Snip1^{Nes}-KO* E13.5 brains. This was a motivation for our RNA-seq analysis of control and *Snip1^{Nes}-KO* differentiating neural (Sox2::GFP-negative) cells, which showed several categories in neuronal specification and differentiation became upregulated in *Snip1^{Nes}-KO* (Supp Fig 5h in original submission and Supp fig 7d in revision). Our collective data suggest that compared with control at E13.5, upregulated apoptosis depletes *Snip1^{Nes}-KO* NPCs, but remnant NPCs give rise to differentiating cells that had upregulated neuronal specification and differentiation. Together, this leads to no significant difference in Tuj1-positive cortical thickness between control and *Snip1^{Nes}-KO*. We apologize that in the initial submission, we did not succeed in articulating our results in the text to make this conceptual connection.

To improve the manuscript, we moved the Tuj1 results from the original Fig 1j-k to revised Supp Fig 7a-b and changed lines 154-172: “Because at E13.5, the embryonic brain undergoes neurogenesis, we examined the immature neuron marker TUJ1. The relative thickness of the TUJ1-positive region did not significantly differ between *Snip1^{Nes}-KO* and control at E13.5 (Supp Fig 7a-b). Considering that *Snip1^{Nes}-KO* NPCs were progressively depleted, this lack of difference was a surprise. Therefore, we aimed to examine the molecular effect of SNIP1 in differentiating neural cells, which have lost the expression of NPC marker SOX2. We performed RNA-seq of SOX2:GFP-negative cells sorted from E13.5 *Snip1^{Nes}-KO* and sibling control brains (Fig 2a, Supp Fig 4a). Using the criteria of fold-change >2 and $p < 0.05$ to compare 2-replicate datasets each from control and *Snip1^{Nes}-KO*, we identified 658 upregulated genes and 150 downregulated genes in *Snip1^{Nes}-KO* (Supp Fig 7c). GSEA revealed that upregulated genes in *Snip1^{Nes}-KO* cells were enriched in functions related to apoptotic clearance, neuronal specification and differentiation, midbrain markers, and known high-CpG-density promoters occupied by bivalent marks (H3K27me3 and H3K4me3) in NPCs⁵⁰ (Supp Fig 7d). Downregulated genes in *Snip1^{Nes}-KO* cells were enriched in functions related to spliceosome, translation and ribosome, nucleosome organization, and apoptosis via p21 but not p53 (Supp Fig 7e). These results suggest that SNIP1 suppresses apoptosis, neuronal specification and differentiation, midbrain genetic programs, and H3K27me3-occupied genes. At E13.5, although upregulated apoptosis reduces *Snip1^{Nes}-KO* NPCs and intermediate progenitors, the remnant of which give rise to cells that had upregulated neuronal specification and differentiation. These in combination likely lead to no apparent difference in TUJ1-positive cortical thickness between *Snip1^{Nes}-KO* and control.”

9. Actually, EED cKO in forebrain also leads to high apoptosis (PMID: 35931079), why *snip1^{Nes}-Eed^{Nes}-Dko* could rescue apoptosis?

Thank you for pointing out this important issue. Zhang et al. (PMID: 35931079) used *Emx1::Cre* (JAX Stock 005628), which is expressed in the dorsal telencephalon starting at E9.5 (Zhou et al Cerebral Cortex, 2002). In contrast, we used *Nes::Cre*, which is expressed throughout the brain starting at E10.5 (Sclafani et al, Genesis, 2006). We crossed *Eed^{fl/fl}* (JAX Stock 022727) with *Nes::Cre* (JAX Stock 003771) to generate *Eed^{Nes}-KO* embryos, which did not show increased apoptosis in ventricular zones when compared to control embryos at E13.5 (now Supp Fig 12d).

We speculate that the function of EED may differ between E9.5 and E10.5 and/or between dorsal telencephalon and other brain regions to contribute to the differences between Zhang et al. and our studies. This is also supported by our observations that apoptosis indeed is readily detectable in brain regions outside of ventricular zones of *Eed^{Nes}-KO* E13.5 brains (white circled areas in Supp Fig 12c-d).

We have added to lines 292-296 “A role of EED in apoptosis was investigated in the control and *Eed^{Emx1}-KO* dorsal telencephalon⁵⁷, which differed from our observations in *Eed^{Nes}-KO* E13.5

brain. This difference could be explained by the expression of *Emx1::Cre* and *Nes::Cre* at different developmental stages and brain regions. Apoptosis was observed in other brain regions of control and *Eed^{Nes}-KO* brains (Supp Fig 12c-d)."

We were also surprised that there was a reduction in apoptosis in *Snip1^{Nes}-Eed^{Nes}-dKO* compared with *Snip1^{Nes}-KO*. Data from Figure 6 and related Supp figures suggest that PRC2 promotes apoptosis in *Snip1^{Nes}-KO*.

Supp Fig 12 c-d. IF of SOX2 and CC3 in sagittal cryosections of the E13.5 brain. Germinal zones around (c) lateral ventricle (Lv) and (d) 3rd /4th ventricles were examined. Bar, 500 μ m.

10. The full name of *Snip1* should be given when it first appears in the paper. And the same is true for other genes.

We thank you for this advice. In the revised manuscript, we give the full name of the genes that are within the focus of the studies. We hope the Reviewer will agree with our rationale for promoting high readability of the manuscript and not providing the full gene names in the following categories. 1) Genes only appear in the manuscript once. 2) Differentially expressed genes and genes appear in the gene set enrichment analysis terms. 3) Immunofluorescence markers. 4) Genes appeared in the motif analysis of CUT&RUN and representative genes for visualizing CUT&RUN data.

11. The expression of the gene SNIP1 is not uniform in case. For example, line 57 is "Snip1", and line 57 is "SNIP1". Please check the names of other genes.

Thank you for pointing this out. We have now changed our manuscript to strictly follow the gene nomenclature system instructed by the Nature Communications, which specifies www.informatics.jax.org/mgihome/nomen. From this guideline, human and mouse proteins are now in all upper case. *Xenopus* and zebrafish proteins have the initial letter capitalized. Human genes are in all italicized upper case and mouse genes are italicized with the capitalized initial letter.

12. Line 254-258, the 32 upregulated genes whether include the 9 genes (Supplementary Figure 7j)?

This is a great point. Of the 9 genes in the original Supplementary Figure 7j, *Cdkn1a* is among the 32 rescued by *Snip1^{Nes}-Eed^{Nes}-dKO*. This suggests that *Snip1* and PRC2 directly alter H3K27me3 at the *Cdkn1a* locus in *Snip1^{Nes}-KO* and *Snip1^{Nes}-Eed^{Nes}-dKO*. We have now this table to Supp Fig 13c and revised lines 317-319 "Of these 32 genes, *Cdkn1a* is the only intrinsic apoptosis gene with significantly lower H3K27me3 levels in *Snip1^{Nes}-KO* versus control (FDR<0.05; Supp Fig 13c), suggesting that SNIP1 directly promotes PRC2 and H3K27me3 at the *Cdkn1* locus."

Supp Fig 13c. List of upregulated intrinsic apoptosis genes with reduced H3K27me3 in *Snip1^{Nes}-KO* NPCs corresponding to

c

Gene	log ₂ (FC)	p-value	FDR
Cdkn1a	-0.95	2.79e-4	0.037
Ppp1r13b	-0.72	5.98e-3	0.091
Trp73	-1.59	8.49e-3	0.104
Hic1	-0.81	0.097	0.318
Msx1	-0.84	0.234	0.507
Tmem117	0.39	0.361	0.631
Sfn	-0.35	0.418	0.679
Aen	-0.15	0.465	0.715
Epha2	-0.04	0.889	0.940

*log₂ (FC) [*Snip1Eed^{Nes}dKO*/*Snip1^{Nes}KO*]

Supp Fig 7j. Fold change, p-value, and FDR of each gene comparing *Snip1^{Nes}-KO* vs. *Snip1^{Nes}-Eed^{Nes}-dKO* are shown.

There are many representative figures are not enough for the journal:

13. Supplementary Fig. 10C: IF analysis of Sox2 and CC3 in E13.5 brain in this figure. However, in this diagram, I don't find staining result of CC3.

We appreciate this opportunity to clarify our results. We did not detect CC3 in control and *Eed^{Nes}-KO* ventricular zones and subventricular zones. Outside these neuroepithelia lined by NPCs, we indeed could detect CC3-positive cells. We added Supp Fig 12c-d, which show that control and *Eed^{Nes}-KO* brains had apoptotic cells.

Supp Fig 12 c-d. IF of SOX2 and CC3 in sagittal cryosections of the E13.5 brain. Germinal zones around (c) lateral ventricle (Lv) and (d) 3rd /4th ventricles were examined. Bar, 500 μ m.

14. Supplementary Figure 2 e and g, why the representative picture is not consistent in the same position?

Thank you for the helpful comment to improve our image representation. We had replaced the images, below, so that the positions are relatively similar between Supp Fig 2e and g.

Supp Fig 2. e, g IF of SOX2 and (e) Ki67 or (g) BrdU in DAPI-stained sagittal cryosections of the E13.5 brain. Germinal zones around the lateral and 3rd ventricles were examined. Bar, 50 μ m. **f, h** Quantification of BrdU-positive, proliferating NPCs in control and *Snip1^{Nes}-*

KO embryos at E13.5. The plot compares one sibling pair, and each data point represents one image. Data are presented as mean \pm SEM, and two-way ANOVA was used for statistical analysis. ns = not statistically significant.

15. Figure 1 h-i ,Supplementary Figure 3 f-h, Supplementary Figure 4 e-g, Supplementary Figure 5a , the statistics of the immunostaining results need to be provided.

We have added statistics that analyzed the immunofluorescence results. Below, we use bolded text to indicate the IF images in the original submission and parentheses to indicate the quantification results / graphs in the revised manuscript. For example, quantification data of the original Fig 1h-i were inserted in Fig 1i, k in the revision.

Fig 1h (Fig 1i)

Fig 1i (Fig 1k)

Here, we clarify that the original Supp Fig 3h is of neuroepithelia within the lateral ventricles of 3g, which lies within 3f. The image resolution within Supp Fig 3g-f was too low to enable rigorous quantification. We quantified images from 3 E13.5 embryos to generate Supp Fig 3i-k for neuroepithelia represented by **Supp Fig 3h**.

(Supp Fig 3i)

(Supp Fig 3j)

(Supp Fig 3k)

Supp Fig 4 e-f (Supp Fig 5d-e) We could not detect γ H2Ax signals through different trials. As γ H2Ax IF signals showed in other samples/tissues, we decided nevertheless show quantification results.

Original **Supp Fig 5a** is now in Supp Fig 6a (quantification in Supp Fig 6b)

16. In figure 5f, the DAPI exposure values of the two groups were inconsistent.

Thank you. We have made sure that the DAPI signals in all images in Figure 5 were consistent.

17. In Supplementary figure 4h, the location of the brain appeared to be inconsistent between the control and Snip1NesKO group, and the expression of Otx2 and foxg1 was demonstrated to be different between the two groups.

Thank you for pointing this out. We had replaced the images that had consistent locations between control and *Snip1^{Nes}-KO*. Supp Fig 4h now corresponds to Supp Fig 5j in the revision.

Supp Fig 5j. IF of FOXG1 (a forebrain marker) and OTX2 (a mid/hindbrain marker) in control and *Snip1^{Nes}-KO* brains at E13.5. Bar, 1 mm.

18. Supplementary Fig. 7 Profiling of H3K27me3 and H3K27me3 in control and Snip1Nes-KO NPCs, the second H3K27me3 should be H3K27ac.

Thank you for pointing this out. It has been corrected.

Reviewer #2 (Remarks to the Author):

Here, Matsui et al reveal a role for Snip1 in suppressing intrinsic apoptosis genes and promoting proliferation in NPCs. They show that it cooperates with PRC2 complex and regulates H3K27 mark turnovers. The data has been tested using both in vivo (KO mice) and In vitro models with a range of biochemical and cellular assays. This study provides interesting insights into the regulation of apoptosis during cortical development, a key process during embryogenesis. However, authors need to address several concerns before this study is considered in Nature Communications.

Comments:

1. It would be good to perform BrdU labelling to compare proliferative content in KO versus WT mice.

Yes, we agree that BrdU labeling is important to test whether SNIP1 depletion affects proliferation. Quantification of BrdU-positive cells (Supp Fig 2g-h) confirmed that SNIP1 depletion did not significantly alter NPC proliferation.

Supp Fig 2g-h Quantification of BrdU-positive, proliferating NPCs in control and *Snip1^{Nes}-KO* embryos at E13.5. The plot compares one sibling pair, and each data point represents each image, which are from 1 sibling pair. Data are presented as mean \pm SEM, and two-way ANOVA was used for statistical analysis. ns = not statistically significant.

2. The claim that Snip1 recruits PRC2 needs to be substantiated by:
-Depletion of Snip1 and analysis of the kinetics of PRC2 components

Respectfully, we infer from the data that PRC2 requires SNIP1 to bind to chromatin. We agree that these experiments will improve our understanding of the PRC2–SNIP1 interaction on chromatin. To examine the effect of SNIP1 depletion on the kinetics of PRC2 components, we used *Snip1[flox/flox]* NPCs transduced with lentiviral control and Cre, followed by assays at Day 2 and 3. This enables our assays with with a temporal precision.

First, we examined the expression of PRC2 components by RT-qPCR. At Day 2, SNIP1 transcripts were depleted by about 70% in *Snip1[flox/flox]* NPCs transduced with lentiviral Cre versus control. By Day 3, SNIP1 level was depleted by almost 99% in *Snip1[flox/flox]* NPCs transduced with lentiviral Cre. SNIP1 depletion did not change the transcript levels of the 10 PRC2 components assayed.

Supp Fig 11b. Transcript levels of PRC2 complex components in *Snip1[flox/flox]* NPCs after 1-3 days of treatment with lentiviral empty control. Normalization was to Actb and values at day 1. Data are presented as mean \pm SEM, and unpaired t tests were used for statistical analysis. **p <0.01; ****p <0.0001

Second, we performed SUZ12 and EZH2 CUT&RUN to analyze PRC2 level on chromatin at days 2 and 3 after *Snip1*[flox/flox] NPCs were transduced with lentiviral Cre versus control. By day 2, there was a modest depletion of PRC2 on chromatin (Supp Fig 11c). And we saw a stronger depletion of SUZ12 and EZH2 on chromatin after 3 days (versus 2 days) of SNIP1 depletion.

Supp Fig 11c. Using replicate SUZ12 and EZH2 CUT&RUN data, MA plots compare the level of SUZ12 and EZH2 binding to chromatin after 2 or 3 days of SNIP1 depletion.

To lines 267-273, we added “Next, we used an in vitro assay to analyze the kinetic effect of SNIP1 depletion on PRC2. Compared with control lentivirus, lentiviral Cre transduction of *Snip1*[flox/flox] NPCs depleted SNIP1 transcripts by 70% and 99.9% at second and third day, respectively. This did not alter the transcript level of PRC2 components (Supp Fig 11b). Using EZH2 and SUZ12 CUT&RUN, we observed a strong reduction of PRC2 on chromatin by the third day of SNIP1 depletion (Supp Fig 11c). Together, these data support that PRC2 requires SNIP1 for binding to chromatin.”

-ChIP-ReChIP to confirm their co-occupancy on chromatin.

We had tested ChIP, which did not work with *Snip1* antibodies. Therefore, we performed SNIP1 CUT&RUN, which was validated by *Snip1*^{Nes}-KO NPCs in the original submission (Fig 3a), and CUT&RUN-reChIP to assay protein co-binding on chromatin. To lines 259-264, we added “To better test PRC2–SNIP1 interactions on chromatin, we performed CUT&RUN-reChIP. In this assay, chromatin

released by SNIP1 CUT&RUN were immunoprecipitated by IgG, EZH2, or H3K27me3. We found that *Mcm7*, *Aen*, *Lhx8*, *Eomes* had co-occupancy of SNIP1 with EZH2 and H3K27me3 but not negative control IgG (Fig 5f). Genome wide, CUT&RUN-reChIP showed EZH2 and H3K27me3 had high overlaps with SNIP1 and PRC2 at SNIP1-bound peaks (Fig 5e).”

Fig 5e. Heatmaps aligning the peaks enriched of SNIP1, SUZ12, and EZH2 in NPCs. Peaks from SNIP1 CUT&RUN–reChIP with IgG, EZH2, and H3K27me3 were aligned to SNIP1-bound peaks. Intensity for 5 or 10 kb on either side of 23,188 Snip1-bound peaks are shown. A dark color indicates high intensity and a light color indicates low intensity. **qPCR** analysis of SNIP1 CUT&RUN–reChIP with IgG, EZH2, and H3K27me3. Primers to SNIP1- and PRC2-bound loci *Foxg1*, *Eomes*, and *Lhx8* were used to show high enrichment, but not negative controls intergenic 1 and 2. **IGV / modified Fig 5f.** SNIP1 CUT&RUN–reChIP with IgG, EZH2, and H3K27me3 tracks visualized by Integrative Genomics Viewer (IGV). In Fig 5f, intergenic region was not shown to due to space constraint.

3. Line No 74. Maybe it is Snip1, not Snip11.

Thank you for pointing this out. It has been corrected.

4. In Figure 2h increasing the concentration of Z LEHD-FMK does not show any effect on intrinsic apoptosis inhibition. Could authors explain this?

Yes, we did not see dosage-dependent inhibition of apoptosis by Z-LEHD-FMK. We think that at 1uM (the lowest concentration tested), Z LEHD-FMK already generated the maximum inhibition of cleaved-caspase 9. For the original submission, we performed this experiment 3 times to get consistent results.

Below showed a new experiment that included higher dosages of Z-LEHD-FMK. As shown, increasing the inhibitor concentration to 20uM and 40uM led to an augmented cytotoxic effect. We

added this piece of data to revised Supp Fig 4e and “Z LEHD-FMK appeared to maximally inhibit caspase 9 at 1uM concentration and cause cytotoxicity at higher concentrations of 20 and 40uM (Fig 2h, supp Fig 5b).” to lines 125-127.

Supp Fig 4e. The percentage of cells with active caspase 3 were quantified by FACS analysis of FAM-DEVD-FMK. Caspase 9 inhibitor (Z-LEHD-FMK) was added at different concentrations along with mCherry-Cre lentivirus. Data are presented as mean \pm SEM, and two-way ANOVA was used for statistical analysis. ns = not statistically significant, * $p < 0.05$, *** $p < 0.001$, and **** $p < 0.0001$.

5. What is the intrinsic apoptosis status in in vitro cultured Snip1Nes-KO NPC cells? Can the induction of apoptosis in Snip1Nes-KO NPC cells be reverted in by overexpressing Snip1?

We agree that our data suggest that intrinsic apoptosis is upregulated in cultured SNIP1-depleted NPCs, and that a more direct assessment would be good. We had tried to use antibody targeting cleaved Caspase 9 to directly assay intrinsic apoptosis in the NPCs. However, we could not detect significant differences between control and SNIP1-depleted NPCs in culture; there was a trend toward statistical significance. We apologize that the results are not as expected.

6. Line No 173. Maybe it is Snip1Nes-KO not Snip1Ne-KO.

Thank you for pointing this out. It has been corrected.

7. In the section “PRC2 requires Snip1 for localization to chromatin and H3K27me3 deposition” Co-immunoprecipitation of Snip1 with PRC2 subunits (Jarid2, Suz12, and Ezh2) in Snip1Nes-KO mutants would have been an interesting negative control.

Thank you for your suggestion. We transduced *Snip1*[flox/flox] NPCs with lentivirus expressing Cre to deplete SNIP1. And we performed co-IP with the nuclear extract of the SNIP1-depleted NPCs. Below, we show the loss of SNIP1 protein in the input lane and the absence of immunoprecipitation of SNIP1 or PRC2 subunits by the SNIP1 antibody. We have included this in Supp Fig 8a and “Anti-SNIP1 antibody did not co-immunoprecipitate with PRC2 subunits in *Snip1*-KO NPCs (Supp Fig 10a), supporting specificity of the antibody.” in lines 248-250.

Supp Fig 10a. Co-immunoprecipitation followed by WB to examine the interaction between SNIP1 and PRC2 subunits in SNIP1-depleted NPCs. *Snip1*[flox/flox] NPCs were transduced with lentivirus expressing Cre for depleting SNIP1.

8. One of the key questions is how *Snip1* is specifically targeted to the different classes of target loci?

We agree that this is a key question. Previous studies showed that SNIP1 participates in TGF β and NF κ B signaling pathways. To test whether they alter SNIP1 binding to chromatin, we used 3 inhibitors targeting components in TGF β or NF κ B signaling pathways. Our results are added to Fig 4, and Supp Table 1. We recognize that our results do not detail a mechanism targeting SNIP1 to chromatin, as changes in SNIP1 CUT&RUN signals were positive and negative in NPCs treated by inhibitors (Fig 4j-m). However, because the changes in SNIP1 CUT&RUN were specific We had uncovered 2 key developmental signaling pathways that influence SNIP1 activities.

We added to line 226-242. **“TGF β and NF κ B signaling pathways control SNIP1 localization to chromatin**

What regulates SNIP1 activities? As SNIP1 participates in TGF β ²²⁻²⁵ and NF κ B²⁶⁻³⁰ signaling pathways, we aimed to test whether their inhibition affects apoptosis in SNIP1-depleted NPCs. We tested 3 inhibitors to TGF β or NF κ B signaling pathway components for toxicity to NPCs (Fig 4a-b). At 0.1-0.5 μ M concentrations, K02288 targeting TGF β (but not Galunisertib or LDN-193189) consistently reduced apoptosis in SNIP1-depleted NPCs (Fig 4c-e). In contrast, 2 inhibitors to TGF β signaling and 3 inhibitors to NF κ B signaling increased apoptosis in NPCs (Fig 4f-h). Therefore, we decided to test whether any of the 6 inhibitors alters SNIP1 binding to chromatin. We treated NPCs with DMSO or inhibitors at 0.5 μ M for 3 days and used CUT&RUN to assay SNIP1 localization on chromatin (Fig 4i). Analyzing replicated SNIP1 CUT&RUN data per treatment condition, we used $p < 0.05$ to identify significant and consistent differences in SNIP1 CUT&RUN signals between inhibitor and control treatments. We identified SNIP1 CUT&RUN changes induced by K02288 treatment (Supp Table 1). We also identified significant changes in SNIP1 CUT&RUN at SNIP1-bound genes that overlap in any 2 inhibitor treatments (Supp Table 1). Average profiling of SNIP1 CUT&RUN signals at SNIP1-bound promoters in NPCs treated with different inhibitors confirmed that inhibition to TGF β (Fig 4j-k) and NF κ B (Fig 4l-m) signaling significantly altered SNIP1 binding to promoters. These data suggest that TGF β and NF κ B signaling pathways control SNIP1 binding to specific gene loci in NPCs.”

Fig 4. Inhibitors to TGF β and NF κ B signaling pathways alter NPC survival and SNIP1 binding to chromatin. **a** Inhibitors targeting components in TGF β and NF κ B signaling pathways and their cytotoxicity at different concentrations. **b** Schematic of inhibitor assay. At day 1, WT or *Snip1*[flox/flox] NPCs were treated with inhibitor or DMSO and transduced with lentiviral Cre for SNIP1 depletion.

DFAM-DEVD-FMK for assaying cl-caspase 3 was used for FACS to quantify apoptotic cells at Day 4. **c-h** The percentage of cells with active caspase 3 quantified by FACS. Inhibitors was added at different concentrations along with mCherry-Cre lentivirus. Data are presented as mean \pm SEM, and two-way ANOVA was used for statistical analysis. ns = not statistically significant; **p<0.01; ***p <0.001; ****p <0.0001. **i** Schematic of SNIP1 CUT&RUN with inhibitor treatment. At day 1, NPCs were treated with DMSO control or different inhibitors. Replicate SNIP1 CUT&RUN was performed for each of the 7 treatment at day 4. **j-m** Profile plots comparing the median binding intensity of SNIP1 in NPCs at the SNIP1-bound targets that had significantly (**j, l**) higher or (**k, m**) lower SNIP1 binding in inhibitors versus DMSO control treatment. **n** indicates region numbers. Regions were considered true *Snip1* targets when *Snip1* levels reduced in *Snip1^{Nes}*-KO vs. control NPCs with p<0.05.

9. Could authors rule out whether the actual role of SNIP1 is keeping a balance between proliferating and differentiating progenitors?

We agree with the Reviewer's assessment that SNIP1 may balance the proliferation and differentiation of NPCs. To test this, we performed in vitro proliferation and differentiation assays, which enable us to study cell-intrinsic role of SNIP1 via a temporal precision.

For in vitro proliferation assays, we used lentiviral control and Cre to transduce *Snip1*[flox/flox] NPCs for genetic depletion. Three days after SNIP1 depletion, we monitored cell proliferation by FACS to quantify the mitotic marker, phosphorylated-serine 10 in histone H3, and neural stem cell marker, SOX2. These analyses showed no difference between control and SNIP1-depleted NPCs. We added to lines 146-151, "We performed in vitro assay to study cell-intrinsic role of SNIP1 for NPC proliferation. *Snip1*[flox/flox] NPCs were transduced with lentiviral control and Cre, and cell proliferation was quantified by FACS of the mitotic marker, phosphorylated-serine 10 in histone H3, and NPC marker, SOX2. This analysis revealed no difference between control and SNIP1-depleted NPCs (not shown). Overall, SNIP1 has a minor role in NPC self-renewal and a major role in suppressing intrinsic apoptosis."

In vitro differentiation assays were initiated at the same day as lentiviral control and Cre transduction. We then used RT-qPCR to assay neuronal and glial differentiation markers at days 1 and 5 of differentiation. Two replicate analyses consistently showed significant upregulation of neuronal and glial markers, suggesting upregulated differentiation. These results are consistent with RNA-seq analysis of SOX2:GFP-negative differentiating cells. Altogether, our collective results from original and revised submissions suggest that SNIP1 has less influence over proliferation but is required for suppressing differentiation of NPCs.

To lines 173-178, we added "Next, we performed in vitro differentiation assay to test cell-intrinsic role of SNIP1 in NPC differentiation. We transduced *Snip1*[flox/flox] NPCs with lentiviral control and Cre at the same day of initiating in vitro differentiation (Supp Fig 7f). At days 1 and 5 of differentiation and compared to control, we observed upregulation of neuronal and glial markers but no difference in NPC markers in SNIP1-depleted cells (Supp Fig 7g-h). Our collective data suggest that SNIP1 suppresses neurogenesis in NPCs."

Supp Fig 7 RT-qPCR profiling of neuronal and glial markers at Day (g) 1 or (h) 5 of in vitro differentiation.

10. Is there any overlap between the genes identified in Suz12 and Ezh2 CUT & RUN assay and the genes that were downregulated in Snip1Nes-KO mutants? If there is minimal or no overlap between these two datasets, it will suggest that Snip1 binds independently of PRC2 at neurodevelopmental gene loci and in combination with PRC2 at intrinsic apoptosis pathway gene loci.

We agree that this is an important point. Of the 1621 downregulated genes in *Snip1^{Nes}-cKO*, 1093 were SNIP1-bound only in WT, 172 genes were bound by both SNIP1 and SUZ12 in WT, and 7 genes were SUZ12-bound only in WT. We performed gene ontology analysis of the 1093 genes bound by SNIP1 only and found significant enrichment in pathways listed below (Supp Fig 8e, left). These data suggest that independent of PRC2, SNIP1 binds to promote the expression of genes involved in G2-M checkpoint, E2F targets, mitotic spindle, mTORC1 signaling, WNT signaling, Hedgehog signaling, and apoptosis. Some of these genes regulate brain development. There were 172 downregulated genes in *Snip1^{Nes}-cKO* that were also bound by both SNIP1 and SUZ12 in WT. This overlap is very little and had few enriched pathways (Below, right graph). Of note, 18 genes were in the apoptosis enriched terms with $p=3.11e-3$; some of these 18 genes might be anti-apoptotic in NPCs.

As predicted by the Reviewer, these results suggest that SNIP1 functions separately from PRC2 to promote genetic programs including neurodevelopment. We added to lines 207-210, “Of the 1621 downregulated genes in *Snip1^{Nes}-KO*, 1093 were SNIP1-bound (not PRC2-bound) and enriched in genetic programs in G2-M checkpoint, cell cycle, key signaling pathways for neurodevelopment, and apoptosis (Supp Fig 8e), suggesting that SNIP1 promotes their expression.”

Left, **Supp Fig 8e**. Gene ontology of the 1093 genes downregulated in *Snip1^{Nes}*-cKO and bound by SNIP1 only in WT. Genes were searched against gene ontology resources using Enrichr⁵⁶.

Right, Gene ontology of the 172 genes downregulated in *Snip1^{Nes}*-cKO and bound by SNIP1 and SUZ12 in WT.

11. Have authors explored whether any SNPs exist in SNIP1 in disease databases in particular for brain disorders?

We agree that this point is indeed important. We followed this suggestion and summarized data from public databases in Supp Fig 1a and added to lines 62-66 “As the E366G variant of *SNIP1* is linked to epilepsy-like neurodevelopmental disorders^{36,37}, we examined publicly available data from Human Gene Mutation Database³⁸ and Mastermind³⁹ and identified a *SNIP1* variant, R111C, that is significantly associated with epilepsy and skull dysplasia (Supp Fig 1a). These data suggest a key role of SNIP1 in human neurodevelopment.”

Supp Fig 1a. Graphical summary of *SNIP1* variants that are associated with epilepsy and skull dysplasia. PMID’s of the source data are listed for the unpublished variants. The R111C variant passed statistical threshold, whereas the other 3 variants nearly passed statistical threshold.

12. There is a lack of data/observations to support the statement at lines 28-29

We agree and have deleted the statement.

13. Authors may investigate and include the detailed observations on intrinsic properties mentioned in Bernard C, Exposito-Alonso D, Selten M, et al. Cortical wiring by synapse type-specific control of local protein synthesis. *Science*. 2022;378(6622):eabm7466. doi:10.1126/science.abm7466

We appreciate the reviewer’s suggestion to help us better our thinking and discussion on intrinsic properties of NPCs. We have now improved lines 356-361 in the Discussion section. “Beyond embryogenesis, *Snip1*–PRC2 may regulate other intrinsic properties of NPCs and neurons. One example is cell type-specific control of synapse formation and neuronal wiring, which is essential to neural circuitry and function⁶⁶. Current and future findings about *Snip1*–PRC2 interactions in NPC survival and maturing neurons will improve our understanding about neurodevelopmental disorders caused by *SNIP1* mutations and PRC2 dysfunction.”

REVIEWER COMMENTS

Reviewer #1 (Remarks to the Author):

I read the revised manuscript and responses to reviewers carefully. The revised version has improved a lot. The authors have addressed my concerns and added the relevant experiments and replaced the low-resolution images. here, I still have several concerns.

1. How about Sox2 or Tbr2 and CC3 colocalization in the ventral zone in Fig1? Have you calculated the ratio of the Sox2+CC3+/Sox2 cells?

2. Actually, the authors added a lot of data to confirm the correlation between PRC2-SNIP1 and it is indeed convincing. I have a concern, does the observation that EED rescue the apoptosis phenotype caused by Snip1 deletion, is it stage specific or only during embryonic stage? In the published research (PMID: 29415247), no apoptosis is shown.

3. The authors found the Tbr2+ cells or apoptosis cells in the Snip1Nes-EedNes-dKO mice could be rescued In Fig.6, is this only shown in E13.5 stage? Have you seen this in E15.5 or other embryonic stages?

4. In already published researches (PMID: 29415247, PMID: 35931079), EED is indeed involved in NSPCs. Here the phenotype in Snip1Nes-EedNes-dKO shows discrepancy, can you find clues from the targets?

Reviewer #2 (Remarks to the Author):

The authors have satisfactorily addressed the majority of my concerns. I recommend publication pending that all the new data is properly discussed in the discussion section.

We appreciate the reviewers for their thoughtful comments to further improve this manuscript. Text changes in the revised manuscript were tracked.

Reviewer #1 (Remarks to the Author):

I read the revised manuscript and responses to reviewers carefully. The revised version has improved a lot. The authors have addressed my concerns and added the relevant experiments and replaced the low-resolution images. here, I still have several concerns.

We appreciate the Reviewer's guidance to further improve the manuscript.

1. How about Sox2 or Tbr2 and CC3 colocalization in the ventral zone in Fig1? Have you calculated the ratio of the Sox2+CC3+/Sox2 cells?

This is a good point. We now have added the quantification of SOX2+CC3+/SOX2 and TBR2+CC3+/TBR2+ in **Supp Fig 2i-l**. These quantifications showed that compared to WT, SOX2+ and TBR2+ cells had significantly increased CC3+ in *Snip1^{Nes}-KO*.

Supp Fig 2i-l Quantification of cell populations that are double-positive for SOX2 and CC3 (i,k) or TBR2 and CC3 (j,l). Each data point represents one image (i,j) or one embryo (k,l). Data are presented as mean ± SEM, and unpaired t-tests were used for statistical analysis. ns = not statistically significant; **p < 0.01; ***p < 0.001; ****p < 0.0001.

2. Actually, the authors added a lot of data to confirm the correlation between PRC2–SNIP1 and it is indeed convincing. I have a concern, does the observation that EED rescue the apoptosis phenotype caused by *Snip1* deletion, is it stage specific or only during embryonic stage? In the published research (PMID: 29415247), no apoptosis is shown. 3. The authors found the *Tbr2*+ cells or apoptosis cells in the *Snip1^{Nes}-Eed^{Nes}-dKO* mice could be rescued In Fig.6, is this only shown in E13.5 stage? Have you seen this in E15.5 or other embryonic stages?

We agree that these are interesting questions and hope that the Reviewer agree with our decision to group questions in points 2 and 3 together. In order to address these questions, we would need to set up multiple timed mating crosses of *Snip1-fl/WT*; *Eed-fl/WT*; *Nes::Cre* parents to obtain and analyze

older embryos and neonates. Because the first set of reviews did not ask about new developmental analyses of *Snip1^{Nes}-Eed^{Nes}-dKO*, we kept one *Snip1-fl/WT; Eed-fl/WT; Nes::Cre* mating cross to maintain the colony for humane and economic reasons,. Since receiving the reviews, we are making crosses to generate more sexually mature *Snip1-fl/WT; Eed-fl/WT; Nes::Cre*. It is a process that takes months. By the 1-month deadline for revising this manuscript, we have not obtained any *Snip1^{Nes}-Eed^{Nes}-dKO* embryos.

We therefore used an alternative strategy to address these questions: new in vitro differentiation (**panels 1-2**) to analyze the interaction of SNIP1 and EED on cultured NPCs and NPC differentiation. After testing different methods, we determined that siRNAs-mediated depletion of SNIP1 results in apoptosis but less robustly so some cells survived and completed the 14-day differentiation course. After 14 days of differentiation, we used RT-qPCR to assay the expression of NPC marker *Sox2*, intermediate progenitor marker *Tbr2*, immature neuronal markers *NeuroD1*, *Dcx*, and *Tuj1/Tubb3*, mature neuronal markers *Satb2* and *Eno2*, and glial markers *Olig1* and *Olig2*. EED depletion in SNIP1-depleted cells results in lower expression of immature neuronal markers but significantly higher expression of mature neuronal markers *Satb2* and *Eno2* (**panel 3**). Glial markers *Olig1* and *Olig2* also increased in SNIP1-EED co-depletion versus SNIP1 depletion (**panel 3**). We conclude that the PRC2–SNIP1 interaction is required for the appropriate expression of neuronal and glial programs in differentiating cells.

We have not included these results in the manuscript because the data do not strongly relate to the rest of the manuscript. The consistency between mature neuronal markers and glial markers suggest that they require PRC2–SNIP1 for appropriate expression during in vitro differentiation.

Fig legends: (1) Schematic of in vitro neural differentiation assay. Wildtype NPCs were transfected with siRNAs targeting *Snip1*, *Eed*, or both on Day 1. To ensure continuous genetic knockdown, cells were transfected with siRNAs 2 more times (on Day 5 and Day 11) before harvested on Day 14. (2) Quantitative PCR of *Eed* and *Snip1* transcripts. (3) Quantitative PCR of neuronal and glial markers on Day 14 of differentiation. The Cq values of each gene were normalized to those of a housekeeping gene *Act*. The expression level of each gene is relative to the level in the control NPCs. Data are presented as mean \pm SEM, and unpaired t-tests were used for statistical analysis. ns = not statistically significant; *p <0.05; **p <0.01; ***p <0.001.

4. In already published researches (PMID: 29415247, PMID: 35931079), EED is indeed involved in NSPCs. Here the phenotype in *Snip1^{Nes}-Eed^{Nes}-dKO* shows discrepancy, can you find clues from the targets?

We appreciate the Reviewer's guidance to better explain our findings in the **"PRC2 promotes apoptosis in the absence of SNIP1"** section.

First, we would like to discuss the 2 published papers. In Sun et al. (PMID 29415247), the effect of PRC2 on neural stem cells were analyzed at P6, ~P14, and P24. The authors showed the requirement of EED for postnatal neural stem cell maintenance and neurogenesis. Embryonic and postnatal neural stem cells have distinctly separate metabolic, molecular, and cellular requirements for maintenance and differentiation. Respectfully, we believe that Sun et al. and our results cannot be directly compared and are not in conflict.

Zhang et al. (PMID 35931079) showed that *Eed^{Emx1}*-cKO increased neurogenesis by 20% at E14.5 and no effect at E16.5 and that *Eed^{Emx1}*-cKO reduced NPC proliferation at E14.5 and E16.5. We did not perform deep analysis of *Eed^{Nes}*-KO, and we only found that it did not show strong induction of apoptosis at E13.5. Our analyses of SNIP1–PRC2 interaction in the brain and cultured NPCs suggest that they are key to neuronal and glial differentiation. Therefore, all 3 studies support that EED/PRC2 regulates neural stem cell survival, maintenance, and differentiation via cell type- and developmental stage-dependency. This is not too surprising as PRC2 dysfunction can be oncogenic or tumor suppressive in a cancer cell-dependent manner. We added to lines 300-302, "As EED has already been shown to affect cortical progenitors at E14.5 but not E16.5 [PMID 35931079], the exact effect of EED/PRC2 on NPC functions is developmental stage- and cell context-dependent."

Second, we are happy about the opportunity to improve this section by discussing the targets. While being mindful of balancing scientific discussion and speculation, we decided to discuss genes that already have known key roles in brain development, apoptosis, or cell cycle progression. We added to lines 338-344, "Indeed, some of the rescued genes by *Snip1^{Nes}-Eed^{Nes}*-dKO versus *Snip1^{Nes}*-KO have high relevance. CDKN1B/P27 and CDKN1A/P21 critically regulate cell cycle progression⁵⁸⁻⁶¹ and apoptosis⁶²⁻⁶⁵. CDC25B⁶⁶⁻⁶⁸, CDK4⁶⁹⁻⁷¹, and PTPN14⁷² are also key to cell cycle progression. HDAC2 regulates transcription and brain development^{73,74}, and *SMS* mutations are causally linked to neurodevelopmental defects in Snyder-Robinson Syndrome⁷⁵⁻⁷⁷. The PRC2–SNIP1 interaction regulates these and other potentially crucial genes in cell cycle progression, apoptosis, or brain development."

Reviewer #2 (Remarks to the Author):

The authors have satisfactorily addressed the majority of my concerns. I recommend publication pending that all the new data is properly discussed in the discussion section.

We thank the Reviewer for the encouraging response and guidance to further improve the manuscript. We added substantial changes to abstract and the discussion sections to properly discuss the new data.

REVIEWERS' COMMENTS

Reviewer #1 (Remarks to the Author):

The authors have addressed my concerns and added the relevant experiments. I recommend publication.

REVIEWERS' COMMENTS

Reviewer #1 (Remarks to the Author):

The authors have addressed my concerns and added the relevant experiments. I recommend publication.

We appreciate the reviewers for their thoughtful comments and support to much improve this manuscript overall.